# Estimation of trace gas fluxes with objectively determined basis functions using reversible jump Markov Chain Monte Carlo

Mark F. Lunt[1], Matt Rigby[1], Anita L. Ganesan[2], and Alistair J. Manning[3]

[1]School of Chemistry, University of Bristol, Bristol, United Kingdom
[2]School of Geographical Sciences, University of Bristol, Bristol, United Kingdom
[3]Hadley Centre, Met Office, Exeter, United Kingdom

*Correspondence to:* Mark Lunt (mark.lunt@bristol.ac.uk)

**Abstract.**

Atmospheric trace gas inversions often attempt to attribute fluxes to a high dimensional grid using observations. To make this problem computationally feasible, and to reduce the degree of under-determination, some form of dimension reduction is usually performed. Here, we present an objective method for reducing the spatial dimension of the parameter space in atmospheric trace gas inversions. In addition to solving for a set of unknowns that govern emissions of a trace gas, we set out a framework that considers the number of unknowns to itself be an unknown. We rely on the the well-established reversible jump Markov chain Monte Carlo algorithm to use the data to determine the dimension of the parameter space. This framework provides a single-step process that solves for both the resolution of the inversion grid, as well as the magnitude of fluxes from this grid. Therefore, the uncertainty that surrounds the choice of aggregation is accounted for in the posterior parameter distribution. The posterior distribution of this transdimensional Markov chain provides a naturally smoothed solution, formed from an ensemble of coarser partitions of the spatial domain. We describe the form of the reversible-jump algorithm and how it may be applied to trace gas inversions. We build the system into a hierarchical Bayesian framework in which other unknown factors, such as the magnitude of the model uncertainty, can also be explored. A pseudo-data example is used to show the usefulness of this approach when compared to a subjectively chosen partitioning of a spatial domain. An inversion using real data is also shown to illustrate the scales at which the data allows methane emissions over north-west Europe to be resolved.

## 1 Introduction

Emissions of atmospheric trace gases can be estimated using observations and an atmospheric chemical transport model (CTM). A common approach to such "inverse" problems uses Bayes theorem; where a prior estimate of parameters, $\mathbf{x}$, is updated by incorporating additional information from observational data, $\mathbf{y}$, based on some set of probability density functions (PDFs), $\rho$, as shown in Eq. (1):

$$\rho(\mathbf{x} \mid \mathbf{y}) = \frac{\rho(\mathbf{y} \mid \mathbf{x}) \cdot \rho(\mathbf{x})}{\rho(\mathbf{y})}. \tag{1}$$

The relationship between the observations and parameters can be determined by a CTM. For flux estimation problems this forward model is usually given by the linear relationship:

$$\mathbf{y} = \mathbf{Hx} + \boldsymbol{\epsilon}. \tag{2}$$

Where $\mathbf{H}$ is a matrix of sensitivities of the observed mole fractions to a change in emissions from a finite grid, calculated by the CTM, and $\boldsymbol{\epsilon}$ represents random representation errors of the observations. The spatial resolution of the CTM output in effect governs a maximum number of spatial parameters for which it is possible to solve (i.e. the number of model grid cells). However, in practice solving at this native resolution can be impractical because: a) there are usually too many unknowns for the number and density of measurement locations; and b) it can be extremely expensive computationally. Instead, the spatial component of the inversion domain may be partitioned into a set of basis functions, where each one represents an element of the parameters vector, $\mathbf{x}$. In addition to the spatial partitioning, some form of temporal aggregation must also be performed, over which the parameters are assumed constant. Each basis function then represents a 3-D aggregation of the underlying fluxes. However, in this work we choose to focus only on the 2-D spatial component of emissions, making the assumption that the fluxes are constant over a fixed period of time.

Traditionally, the number of basis functions has been fixed a priori, and the flux values associated with each basis function are updated in the inversion. The basis functions can take various forms, but often each one represents a 2-D geographical area, within which the fluxes are either uniform (e.g. Manning et al., 2011) or fixed according to some prior distribution (e.g. Rigby et al., 2011; Lunt et al., 2015). Either way, the designation of these basis functions enforces a correlation on emission errors within that area, although adjacent regions are often treated as uncorrelated. The choice of how many basis functions for which to solve can be seen as a balance between an under-determination of sources and so-called aggregation error, which is used to describe an error brought about by allowing the model too few degrees of freedom (Kaminski et al., 2001; Turner and Jacob, 2015). If $\mathbf{x}$ contains a very large number of elements, there may not be enough information in the data to accurately constrain each element. If the prior is poorly specified, this could lead to the solution being overly influenced by the incorrect prior, leading to a so-called smoothing error (e.g. von Clarmann, 2014). If the error in the prior is low, then, while a greater number or degrees of freedom would improve the ability to fit the data, the computational expense of such a calculation may be critical (Bocquet et al., 2011). Conversely, if there are too few elements of $\mathbf{x}$, then solutions may exist which hide the true spatial flux patterns within large aggregated regions, creating aggregation error (Kaminski et al., 2001).

Different methods can be used for this dimension reduction such as radial basis functions or principal components (Turner and Jacob, 2015), but it is perhaps more commonly based on a spatial aggregation of neighbouring or correlated grid-cells. Many subjective approaches have been taken, such as decreasing the basis function resolution with increasing distance from measurement sites (e.g. Manning et al., 2011; Brunner et al., 2012), or by considering the areas which give rise to the largest mole fraction enhancements seen at measurement sites (Rigby et al., 2011). Thompson and Stohl (2014) adopted a similar approach, where a coarse grid was chosen as the starting point, and only refined in those areas where the flux sensitivity was above a certain threshold. However, whilst care has been taken in each of these studies to select a suitable configuration of basis

functions, there is no guarantee that this a priori choice is the most appropriate for a given dataset. Furthermore, the uncertainty surrounding the choice of basis functions does not propagate through to the posterior flux estimate.

There are only a few studies that have sought to solve for the partitioning of basis functions, such that the aggregation of the parameter space may be performed objectively. Bocquet (2009) proposed an adaptive grid system to optimize the basis functions for the inversion, dependent on the average posterior reduction in uncertainty. A further method developed in Bocquet et al. (2011) and Wu et al. (2011) considered solving for the maximum degrees of freedom for the signal, as the basis for an optimum partitioning of the inversion domain. More recently, Turner and Jacob (2015) proposed a method to determine the optimum dimension of the parameters vector, by minimising the combination of aggregation and smoothing error. Although a parameter dimension was successfully identified which minimised the total error, ultimately the choice of model to use was as much influenced by computational efficiency, as it was by this combination of aggregation and smoothing error. This follows the work of Bocquet et al. (2011) who showed that the highest resolution grid should have the smallest total error, and thus computational efficiency is the main driver behind dimension reduction. Importantly, these methods rely on a two-step process of determining an optimum resolution on which to perform the inversion, followed by inference on the parameters of interest. Therefore, the uncertainties in step one do not necessarily propagate through to step two.

An alternative to proposing an optimum partitioning of basis functions either subjectively or otherwise, is to instead allow the data to decide the form of the partitioning. Such an approach has been receiving increasing attention in many fields of the geosciences which invoke Bayes theorem (Sambridge et al., 2013, and references therein). In this alternative approach, the number of basis functions, and their configuration in the inversion domain, becomes a variable to be solved for in the inversion. Since the dimension of the parameter space can vary, this type of inversion has been termed a transdimensional inversion (Green, 2003).

There are three critical advantages to this transdimensional approach: a) the subjectivity associated with the choice of partitioning is largely removed; b) the uncertainty that surrounds the choice of partitioning is propagated through to the posterior parameters estimate; c) the partitioning of the inversion domain and inference on the desired parameters are calculated simultaneously.

One might expect that such an inversion would attempt to find a better fit with the data simply by prescribing a larger number of basis functions. However, Bayes theorem follows the principle of Occam's razor, in that, given a choice between a complicated and simple solution that provide similar fits to the data, the simpler one will be favoured (Sambridge et al., 2006). The term that controls this feature of Bayesian inference is the denominator of Eq. (1), $\rho(\mathbf{y})$, commonly termed the "evidence" or Bayes factor. This term is usually ignored in Bayesian inferences on $\mathbf{x}$, since it is independent of the parameter values. Thus, it acts merely as a normalising constant, which cancels out when comparing one set of parameter values to another. When the basis function dimension is fixed, it can be considered irrelevant for this first level of inference (MacKay, 2002). However, the evidence is very much relevant when we wish to compare two different sets of hypotheses regarding the choice of basis function model.

For two different models (which contain a different number of basis functions), $\mathbf{m}_1$ and $\mathbf{m}_2$, the evidence can tell us which model is more probable, given the data. Following MacKay (2002) and Sambridge et al. (2006) the posterior probability of each model, $\mathbf{m}_i$, is:

$$\rho(\mathbf{m}_i \mid \mathbf{y}) \propto \rho(\mathbf{y} \mid \mathbf{m}_i) \cdot \rho(\mathbf{m}_i). \tag{3}$$

The first term on the right hand side is the evidence, which appeared as the normalizing constant in Eq. (1). The only difference here is that we acknowledge that the evidence is dependent on the choice of model $\mathbf{m}_i$. The second term is the prior probability we assign to each model. If we assume equal prior weights for any two models, then the posterior probability of model 1 versus model 2 simply becomes the ratio of the conditional evidence of each model. Hence, it is the evidence that determines which model is the more probable.

The evidence gives a measure of the probability of randomly choosing the set of parameter values that generate the data, $\mathbf{y}$. Models that are too simple (too few basis functions) have little chance of recreating $\mathbf{y}$, since there are not enough degrees of freedom. However, models that are very complex (too many basis functions) have a far greater number of possible parameter value combinations, and hence the probability of randomly picking the set of parameters that generate $\mathbf{y}$ is also small. As shown by Eq. (3), the model which has the greater evidence will be the one that has the greater posterior probability, and

accordingly will provide the more likely solution to the inverse problem. This prevents the hypothesis that could best fit the data (i.e. contains the maximum number of unknowns) being that which is the most probable.

Theoretically, one could use the evidence as a means to find the partitioning of the inversion domain that provides the most likely explanation for the given data. However, in practice this is not so straightforward, as the evidence can be particularly complex to calculate, particularly for non-linear, high-dimensional problems. Instead, we can make use of the reversible jump

Markov chain Monte Carlo (rj-MCMC) method of Green (1995). This algorithm is an extension of the Metropolis-Hastings approach (Metropolis et al., 1953; Hastings, 1970; Tarantola, 2005) that has been used previously to estimate fluxes in atmospheric inverse modelling (e.g. Rigby et al., 2011; Miller et al., 2014; Lunt et al., 2015). The rj-MCMC algorithm allows one to propose changes not just to parameter values, but to the parameter dimension as well, and it has been widely used in many fields to address the problem of model selection (Hastie and Green, 2012). It is this rj-MCMC approach that forms the

backbone of the transdimensional inversion.

Sambridge et al. (2006) showed how the evidence and the transdimensional approach are intrinsically linked. Given a relatively simple one-dimensional case, they showed that the posterior distribution of the number of basis functions inferred through using rj-MCMC, replicated the ratio of the conditional evidences for fixed dimension inversions for each number of unknowns. As such, using a transdimensional approach elegantly incorporates the benefits of taking the evidence into account,

without the requirement of actually calculating it.

In addition to being dependent on the partitioning of basis functions, Bayesian inversions are also dependent on the form of the PDFs used to describe the prior and likelihood. The terms that describe these PDFs such as the mean, standard deviation and correlation length are commonly referred to as hyperparameters. The dependence of the posterior parameters on

these hyperparameters, and a lack of objective determination of their values have been previously identified as a limitation of Bayesian inverse methods (e.g. Rayner et al., 1999). There have since been a number of studies that have proposed methods for determining hyperparameter values using the data (e.g. Michalak et al., 2005; Berchet et al., 2013; Wu et al., 2013). In general, these methods rely on Gaussian assumptions and are performed in a two-step process whereby the hyperparameters are first optimised, and then parameter inference is performed based on these optimal values. Winiarek et al. (2012) also extended this to a semi-Gaussian prior PDF, such that the source term was constrained to be positive. However, as noted by Berchet et al. (2015), one issue is that the uncertainty in the specification of the hyperparameters in step one is not propagated through to the second step.

Ganesan et al. (2014) presented an alternative method, where the hyperparameters and parameters were estimated simultaneously using an MCMC algorithm. This framework explored the "uncertainties in the uncertainty", resulting in a more complete characterization of the uncertainty in the posterior parameters. The framework also has the advantage that the data is used only once, thus remaining strictly Bayesian, and PDFs are able to take forms other than Gaussian. In the transdimensional case, the posterior distribution of the number of unknowns can be heavily dependent on the prescribed uncertainties (Bodin et al., 2012). As such, it is important to incorporate data driven hyperparameters into the transdimensional inversion, if the derived number of unknowns is to be truly dependent on the data.

In this work, we will set out a transdimensional hierarchical Bayesian inverse framework and its application to estimating emissions of trace gases, using atmospheric data. We describe the form of the partitioning of the model space, and how this may be easily varied in the inversion. We further incorporate a hierarchical Bayesian framework (Ganesan et al., 2014) to allow the hyperparameters describing the prior and model-measurement uncertainties to themselves become variables in the inversion. The method is applied to an idealised pseudo-data example and compared to a typical fixed-grid inversion. Finally we apply the transdimensional, hierarchical approach to determine methane emissions across North West Europe using data from a UK in-situ greenhouse gas monitoring network. We present this transdimensional approach as an extension of the hierarchical Bayesian framework, and as an alternative to defining a single optimum set of basis functions.

## 2 Methodology

### 2.1 The Basis Functions

Partitioning based on a spatial aggregation of neighbouring cells allows one to easily define natural boundaries such as a land-sea interface or country border, but can also have the effect of imposing hard boundaries elsewhere, which is an extremely crude depiction of reality. Furthermore, each grid cell within each aggregated region has an enforced uncertainty correlation to its neighbours, which may not be appropriate. However, whilst other forms of basis functions may avoid these pitfalls, they can lack the simplicity of a simple grid coarsening. For this reason, the basis functions we define in this work are based on a spatial aggregation of neighbouring cells. However, the same transdimensional framework could be similarly applied to determine the number of appropriate principal components, radial basis functions or indeed any other model reduction class method to use in an inversion.

### 2.1.1 Voronoi Cells

Instead of defining the spatial domain as a series of regular square or rectangular basis functions, an alternative is to assign a number of nodes, or nuclei, to the domain. Every nucleus defines a region, where the edge of each region is equidistant between the closest two nuclei, and perpendicular to a line connecting the nuclei pair. As such, any point within each region is closer to that region's nucleus than any other nucleus. Figure 1 shows how a spatial domain may be partitioned into ten such "Voronoi cells", given a set of 10 nuclei.

On its own, partitioning a domain into Voronoi cells is perhaps a rather unrefined means of forming a set of basis functions, since it does not take account of natural partitions, such as a land-sea boundary, or different vegetation types. However, defining a grid by a set of Voronoi nuclei provides a convenient way of describing the size and shape of each cell using just two values: the x and y coordinates of each nuclei. The ease of describing a change in the partitioning of Voronoi cells means that it is easy to define multiple Voronoi meshes on the same domain, simply by selecting different nuclei.

The basis functions we define represent not just the magnitude of emissions from each Voronoi cell, but the location of each nuclei as well. The set of combined basis functions, or partition model, can be represented by:

$$\mathbf{m} = (\mathbf{c}, \mathbf{x}). \tag{4}$$

Where $\mathbf{c}$ represents the longitude and latitude, and $\mathbf{x}$ is the flux value associated with each Voronoi nucleus. Changing the location of just one of the Voronoi nuclei, will impact on the boundaries of the surrounding Voronoi cells, as shown in Fig. 1. Some of the hard boundaries that exist in the first realisation of the Voronoi cells have moved in the second realisation. The transdimensional inversion involves continuously making changes of this type, as well as adding and removing Voronoi nuclei. For simplicity, in this work we define the form of the Voronoi cells such that they must be composed of whole grid cells of an underlying finite grid. As such the whole of each native resolution grid cell can belong to only one Voronoi region, and grid cells are not split between two or more regions. This approach limits the maximum resolvable resolution to be the same as the underlying grid (defined by the CTM output).

## 2.2 Bayes Theorem for the transdimensional case

The use of Eq. (1) in previous trace gas inversion assumes a fixed dimensionality for $\mathbf{x}$, however this is only tacitly implied. If we ignore the evidence term, then more properly Eq. (1) should be rewritten:

$$\rho(\mathbf{m} \mid \mathbf{y}, k) \propto \rho(\mathbf{y} \mid \mathbf{m}, k) \cdot \rho(\mathbf{m} \mid k). \tag{5}$$

Where $k$ represents the number of unknown parameters, and $\mathbf{x}$ has been replaced by $\mathbf{m}$ to account for the fact that this includes information on the position of each region, as well as the emissions. Equation (5) explicitly states that the posterior PDF of $\mathbf{m}$ is dependent upon the dimensionality of $\mathbf{m}$, i.e. the number of unknowns. In the transdimensional inversion we want to infer information, not just on the model parameters, but on the dimensionality of the model as well. Thus, Eq. (5) must be re-expressed to solve for the joint posterior of the partition model and the number of unknowns.

Using the property of PDFs that states $\rho(a,b) = \rho(a|b) \cdot \rho(b)$, one can express the relationship between the terms as follows:

$$\rho(\mathbf{m}, k \mid \mathbf{y}) = \rho(\mathbf{m} \mid \mathbf{y}, k) \cdot \rho(k \mid \mathbf{y}). \tag{6}$$

Bayes theorem allows the term $\rho(k \mid \mathbf{y})$ to be split into its constituent parts, so that, by combining Eq. (5) and Eq. (6), we are left with Eq. (7), which is Bayes theorem for the transdimensional problem.

$$\rho(\mathbf{m}, k \mid \mathbf{y}) \propto \rho(\mathbf{y} \mid \mathbf{m}, k) \cdot \rho(\mathbf{m} \mid k) \cdot \rho(k). \tag{7}$$

In addition to $\mathbf{m}$ and $k$ we also wish to solve for the set of hyperparameters that describe the prior parameters PDF, $\boldsymbol{\theta_x}$, and the likelihood PDF, $\boldsymbol{\theta_y}$. The dimension of the latter is independent of $k$ since it is a property of the data. However, we prescribe the dimension of the emissions hyperparameters, $\boldsymbol{\theta_x}$, to be dependent on $k$, alongside the parameters, such that each basis function is described by its own unique prior PDF. The full form of the transdimensional, hierarchical Bayesian equation then becomes:

$$\rho(\mathbf{m}, \boldsymbol{\theta_x}, \boldsymbol{\theta_y}, k \mid \mathbf{y}) \propto \rho(\mathbf{y} \mid \mathbf{m}, \boldsymbol{\theta_y}, k) \cdot \rho(\mathbf{m} \mid \boldsymbol{\theta_x}, k) \cdot \rho(\boldsymbol{\theta_x}|k) \cdot \rho(k) \cdot \rho(\boldsymbol{\theta_y}). \tag{8}$$

Ganesan et al. (2014) used the Metropolis-Hastings algorithm to simultaneously solve for $\boldsymbol{\theta_x}$, $\boldsymbol{\theta_y}$ and $\mathbf{x}$ in the hierarchical framework. In the transdimensional inversion we also wish to vary the dimension of $\mathbf{m}$, which can be achieved using the more general rj-MCMC technique described by Green (1995), as set out in section 2.3.

## 2.3 Reversible Jump MCMC

Conventional MCMC techniques allow the flexibility of including models which are non-Gaussian, or with varying hyperparameters, and thus cannot be solved analytically. In this usual Metropolis-Hastings approach, at each iteration a proposal is made to move to a new point in the state-space, and subsequently either accepted or rejected based on some probabilistic criterion. The proposal is accepted provided that:

$$U \leq (\text{prior ratio} \times \text{likelihood ratio}). \tag{9}$$

Where U is an uniformly-distributed random number between 0 and 1. The prior ratio represents the movement away form the a priori parameter values of the current and proposed state. The likelihood ratio is given by the first term on the right hand side in Eq. (7) and represents the match (or mismatch) to the data of the current and proposed states on the chain. Thus, whether a move is accepted or not depends on a balance between the weight of the prior and the data. In addition, there is also a probability (based on $U$) that a state that is not probabilistically favourable will also be accepted.

However, conventional MCMC assumes that the model (the dimension of $\mathbf{m}$) on which the inversion is performed is known, and defined from the start. Equation (7) shows that this no longer applies in the transdimensional case, hence a different method must be used. The reversible-jump algorithm (Green, 1995, 2003) expands on the fixed dimension case, to provide a more general expression for the acceptance term if we wish to consider changes in dimension. Green (1995) showed that a proposal will be accepted for the transdimensional case provided that:

$$U \leq (\text{prior ratio} \times \text{likelihood ratio} \times \text{proposal ratio} \times |\mathbf{J}|), \tag{10}$$

Or more formally:

$$U \leq \left( \frac{\rho(\mathbf{m}')}{\rho(\mathbf{m})} \times \frac{\rho(\mathbf{y}'|\mathbf{m}')}{\rho(\mathbf{y}|\mathbf{m})} \times \frac{q(\mathbf{m}|\mathbf{m}')}{q(\mathbf{m}'|\mathbf{m})} \times |\mathbf{J}| \right). \tag{11}$$

Equation (10) contains two additional terms compared to Eq. (9). The first, $\mathbf{J}$, is a Jacobian matrix that comes into the acceptance ratio on account of the proposal destination $\mathbf{m}'$ being specified through some deterministic function of the current state, $\mathbf{m}$ (Hastie and Green, 2012). For the purposes of this work, we will consider a special case of the reversible jump algorithm, namely birth-death MCMC (Geyer and Moller, 1994). This algorithm considers the case where only dimensional changes that are one more, or one less are allowed. For this special case it can be shown that the term $|\mathbf{J}| = 1$ and so can be conveniently ignored (Bodin and Sambridge, 2009).

The second additional term is the proposal ratio, $\frac{q(\mathbf{m}'|\mathbf{m})}{q(\mathbf{m}|\mathbf{m}')}$. At each step along the chain, we propose a new state for the model, $\mathbf{m}$. In fixed dimension Metropolis-Hastings applications, this is usually performed by selecting a random perturbation from a normal distribution, N, centred on 0 and with standard deviation $\sigma$.

$$\mathbf{m}' = \mathbf{m} + \mathbf{n} \quad \text{where } \mathbf{n} \sim N(0, \sigma). \tag{12}$$

To perform this step, we are in effect applying a proposal distribution, in this case Gaussian, denoted by $q(\mathbf{m}'|\mathbf{m})$, to this change of state, which needs to be taken into consideration. However, the acceptance term depends not on the distribution itself, rather the ratio of the distributions, $\frac{q(\mathbf{m}'|\mathbf{m})}{q(\mathbf{m}|\mathbf{m}')}$. Since the Gaussian proposal distribution is symmetric, $q(\mathbf{m}'|\mathbf{m}) = q(\mathbf{m}|\mathbf{m}')$, the two terms cancel each other out, and the proposal distribution is always 1. For this reason it does not need to be considered for conventional, fixed dimension Metropolis-Hastings. However, for the case of a change of dimension, this symmetry no longer always applies, and so the proposal distributions and ratios must be calculated.

The reversible jump algorithm allows the sampling of arbitrary dimension probability density functions (PDFs), and thus allows us to explore both the model parameter values, and the model dimensionality simultaneously. At each step on the chain we prescribe five possible proposals:

1. Emissions update - randomly select and perturb one emissions value

2. Hyperparameter update - randomly select and perturb one hyperparameter value

3. Move - randomly select and move one Voronoi nucleus location

4. Birth - add one new Voronoi nucleus to a random location in the domain, thereby increasing the parameter space by one

5. Death - randomly select and remove one Voronoi nucleus, thereby reducing the parameter space by one.

The first two steps involve a change only in the emissions value prescribed to a cell, or a hyperparameter value, exactly the same as a conventional fixed-dimension hierarchical inverse problem. The other three proposal types involve a change to the partitioning of the grid, either through a dimension change, or by moving the location of a nucleus. This means that the sensitivity matrix, $\mathbf{H}$, which maps the relationship between emissions and observations, must be recalculated for the new set of aggregated regions. Movement along the chain, due to any of the five possible proposals, is dependent only on the prior, likelihood and proposal ratios. We explore each of these distributions and ratios in detail in the following subsections.

## 2.3.1 Prior distributions

In the transdimensional inversion, there is an unknown number of unknowns, so the prior PDF must describe both the basis functions, $\mathbf{m}$, and the number of unknowns, $k$. The dependency of $\mathbf{m}$ on $k$ is given by:

$$\rho(\mathbf{m}) = \rho(\mathbf{m} \mid k) \cdot \rho(k). \tag{13}$$

Partitioning the inversion domain into Voronoi cells enables us to describe the basis functions using three parameters: the longitude, latitude and emissions value. If the emissions value is taken to be a scaling of the a priori distribution of emissions, then the a priori scaling of this prior emissions field should be one everywhere, and hence this is not dependent on location. In this work we assign a uniform distribution for the location of the Voronoi nuclei, meaning that the prior distribution is independent of the emissions. Given this independence of the variables, the term $\rho(\mathbf{m} \mid k)$ can be decomposed into two terms expressed as:

$$\rho(\mathbf{m}|k) = \rho(\mathbf{c} \mid k) \cdot \rho(\mathbf{x} \mid k). \tag{14}$$

Where $\mathbf{c}$ represents a set of variables that describe the location of the Voronoi nuclei of the basis functions, and $\mathbf{x}$ is the emissions scaling variable. The full prior distribution of the basis functions is therefore:

$$\rho(\mathbf{m}) = \rho(\mathbf{c} \mid k) \cdot \rho(\mathbf{x} \mid k) \cdot \rho(k). \tag{15}$$

Given a uniform distribution, a priori the Voronoi nuclei may be located anywhere within the spatial inversion domain. However, if we assume that the Voronoi nuclei can only be located at the centre points of each grid cell on a finite underlying grid with $K$ grid cells, and that no two nuclei can occupy the same grid cell, then for $k$ Voronoi nuclei there are $\frac{K!}{k!(K-k)!}$

possible configurations. Since we are assuming each position on the grid has equal probability, the prior PDF $\rho(\mathbf{c}|k)$ is given by:

$$\rho(\mathbf{c}|k) = \left[ \frac{K!}{k!(K-k)!} \right]^{-1}. \tag{16}$$

For the number of unknowns, we assume little prior knowledge on this quantity, and assign a uniform distribution that can take any value between a maximum and minimum. Outside of this range the probability is set as zero. Whilst the uniform prior is relatively uninformative, the choice of maximum and minimum bounds may still influence the number of nuclei if the constraint from the data is weak or if the bounds are too narrow.

$$\rho(k) = \frac{1}{(k_{max} - k_{min})} \quad \text{if } k_{min} < k \leqslant k_{max}, 0 \text{ otherwise.} \tag{17}$$

For emissions, we have chosen a lognormal PDF, since it is usually a requirement that anthropogenic emissions are defined only on the positive axis, and gridded emissions databases are readily available. However, in cases where the distribution of emissions is less certain or unknown, one could easily use an alternative PDF, such as an uniform distribution, again defined only on the positive axis. For a lognormal distribution the term $\rho(\mathbf{x} \mid k)$ can be expressed:

$$\rho(\mathbf{x}|k) = \frac{1}{\mathbf{x}\sqrt{|\boldsymbol{\Sigma_x}|(2\pi)^k}} \cdot \exp\left( \frac{-(\ln\mathbf{x} - \boldsymbol{\mu_x})^T \boldsymbol{\Sigma_x}^{-1}(\ln\mathbf{x} - \boldsymbol{\mu_x})}{2} \right). \tag{18}$$

The full prior PDF of the basis functions is thus:

$$\rho(\mathbf{m}) = \left[ \frac{K!}{k!(K-k)!} \right]^{-1} \cdot \frac{1}{(k_{max} - k_{min})} \cdot \frac{1}{\mathbf{x}\sqrt{|\boldsymbol{\Sigma_x}|(2\pi)^k}} \cdot \exp\left( \frac{-(\ln\mathbf{x} - \boldsymbol{\mu_x})^T \boldsymbol{\Sigma_x}^{-1}(\ln\mathbf{x} - \boldsymbol{\mu_x})}{2} \right)$$
$$\text{if } k_{min} < k \leqslant k_{max}, 0 \text{ otherwise.} \tag{19}$$

We assume minimal prior knowledge on the emissions hyperparameters, $\boldsymbol{\sigma_x}$ (which are the square-roots of the diagonals of $\boldsymbol{\Sigma_x}$) and $\boldsymbol{\mu_x}$, except that they may vary between some range of a uniform distribution. Similarly, for the hyperparameters that describe the model-measurement covariance structure, namely the model uncertainty, $\sigma_{\mathbf{y}}$, and a correlation time scale, $\tau$, we use a uniform distribution for the prior.

### 2.3.2 Proposal distributions

For the transdimensional case, we must consider the form of the proposal ratio $\frac{q(\mathbf{m}|\mathbf{m}')}{q(\mathbf{m}'|\mathbf{m})}$ (Green, 1995). The denominator describes the probability of generating a perturbed model, $\mathbf{m}'$ starting from the current model, $\mathbf{m}$. The numerator describes the probability of the reverse process of generating $\mathbf{m}$ from $\mathbf{m}'$. As the name suggests, the form of the proposal distribution is dependent on the type of movement along the chain that is proposed.

As previously discussed, when updating an emissions value of a basis function, the proposal distribution takes the form of a Gaussian perturbation to the current state. Therefore, the proposal distribution is symmetrical, and $\frac{q(\mathbf{m}|\mathbf{m}')}{q(\mathbf{m}'|\mathbf{m})} = 1$. A similar argument applies to a change in one of the hyperparameters. However, this simplicity does not necessarily apply to the other proposal types.

5      The third proposal is to randomly select one Voronoi nucleus and move its location according to some Gaussian distribution, centred on its current position. The emissions value associated with that nucleus remains unchanged. Again, as in the case of the emissions update, this proposal distribution is symmetric, and the proposal ratio equals 1.

The birth proposal involves randomly selecting a vacant point in the domain to add a new Voronoi nucleus. The new nucleus also requires an emissions value, which is chosen based on a Gaussian perturbation of the emissions value of the Voronoi cell, 10    $i$, in which that point currently sits. The new nucleus is generated independently of the new emissions value, so the proposal ratio $\frac{q(\mathbf{m}|\mathbf{m}')}{q(\mathbf{m}'|\mathbf{m})}$ can be split into two independent parts:

$$
\frac{q(\mathbf{m}|\mathbf{m}')}{q(\mathbf{m}'|\mathbf{m})} = \frac{q(\mathbf{c}|\mathbf{m}')}{q(\mathbf{c}'|\mathbf{m})} \cdot \frac{q(\mathbf{x}|\mathbf{m}')}{q(\mathbf{x}'|\mathbf{m})}. \tag{20}
$$

Assuming we have a finite grid with $K$ points, and $k$ current nuclei, then it can be shown (Bodin and Sambridge, 2009) that the proposal ratio for a birth takes the form:

$$
\left[ \frac{q(\mathbf{m}|\mathbf{m}')}{q(\mathbf{m}'|\mathbf{m})} \right]_{birth} = \frac{\sqrt{2\pi}(K-k)}{k+1} \cdot \sigma_{bd} \cdot \exp\left( \frac{(x'_{k+1} - x_i)^2}{2\sigma_{bd}^2} \right). \tag{21}
$$

Where $x'_{k+1}$ is the new emissions value at the new nucleus based on the current value, $x_i$, at the new nuclei location. The size of the Gaussian perturbation from $x_i$ is given by $\sigma_{bd}$.

The death process of removing a Voronoi nucleus is the exact opposite of the birth step of adding a Voronoi nucleus. Supposing that the $k^{th}$ nucleus is removed, along with emissions value $x_k$. The point which is removed would become part of 20    a different Voronoi cell, $j$, and take the emissions value of that cell, $x'_j$. Bodin and Sambridge (2009) showed that the proposal ratio would then take the form:

$$
\left[ \frac{q(\mathbf{m}|\mathbf{m}')}{q(\mathbf{m}'|\mathbf{m})} \right]_{death} = \frac{k}{\sigma_{bd}\sqrt{2\pi}(K-k+1)} \cdot \exp\left( \frac{-(x'_j - x_k)^2}{2\sigma_{bd}^2} \right). \tag{22}
$$

### 2.3.3 Likelihood function

Here, we assume that the likelihood function is based on a least-squares misfit. The form of this function is given by:

25    $\rho(\mathbf{y}|\mathbf{m}, k) = \frac{1}{\sqrt{|\mathbf{R}|2\pi}} \cdot \exp(\frac{-\Phi(\mathbf{m})}{2}).$               (23)

Where $\mathbf{R}$ is the model-measurement covariance matrix and $\Phi(\mathbf{m})$ represents:

$$\Phi(\mathbf{m}) = (\mathbf{y} - \mathbf{Hm})^T \mathbf{R}^{-1} (\mathbf{y} - \mathbf{Hm}). \tag{24}$$

The model-measurement covariance matrix $\mathbf{R}$ can be composed of two hyperparameters; one describing the model-measurement uncertainty, $\boldsymbol{\sigma_y}$, and the other a correlation length between measurement uncertainties, $\tau$. Proposals that involve changes to $\boldsymbol{\sigma_y}$

5   or $\tau$ result in $\mathbf{R}$ also being perturbed, and its inverse and determinant must be recalculated. At first glance this may appear to be a cumbersome or impractical step.

However, careful design of the covariance structure can simplify the problem. The covariance matrix $\mathbf{R}$ can be expressed as the product of a diagonal matrix of standard deviations, $\boldsymbol{\Sigma}$, and a correlation matrix $\mathbf{Q}$:

$$\mathbf{R} = \boldsymbol{\Sigma}\mathbf{Q}\boldsymbol{\Sigma}. \tag{25}$$

10   The inverse may be similarly defined as:

$$\mathbf{R}^{-1} = \boldsymbol{\Sigma}^{-1}\mathbf{Q}^{-1}\boldsymbol{\Sigma}^{-1}. \tag{26}$$

Changes to $\boldsymbol{\sigma_y}$ only change the diagonal matrix $\boldsymbol{\Sigma}$, the inverse of which is simply its reciprocal. Changes to $\tau$ are more complicated since these alter the structure of $\mathbf{Q}$. However, if an exponential covariance form is used, then given a set of data points with a regular time interval, $\delta t$, the correlation matrix can be expressed as a symmetric Toeplitz matrix of the form:

$$15 \quad \mathbf{Q} = \begin{pmatrix} 1 & q & q^2 & \cdots & q^{N-1} \\ q & 1 & q & \cdots & q^{N-2} \\ q^2 & q & 1 & \cdots & q^{N-3} \\ \vdots & \vdots & \vdots & \ddots & \vdots \\ q^{N-1} & q^{N-2} & q^{N-3} & \cdots & 1 \end{pmatrix}. \tag{27}$$

Where $N$ is the number of data points and $q$ can be expressed in terms of the correlation timescale, $\tau$, and $\delta t$ following:

$$q = exp\left(-\frac{\delta t}{\tau}\right). \tag{28}$$

This type of symmetric Toeplitz matrix has an explicit inverse (Malinverno and Briggs, 2004), which requires computation and storage that is proportional to $N$ and is given by:

$$
\mathbf{Q}^{-1} = \frac{1}{1-q^2} \begin{pmatrix} 1 & -q & 0 & \cdots & 0 \\ -q & 1+q^2 & -q & \cdots & 0 \\ 0 & -q & 1+q^2 & \cdots & 0 \\ \vdots & \vdots & \vdots & \ddots & \vdots \\ 0 & 0 & 0 & \cdots & 1 \end{pmatrix}.
\tag{29}
$$

The determinant of $\mathbf{R}$ can also be easily defined as:

5    $det(\mathbf{R}) = \sigma_1^2 \sigma_2^2 \cdots \sigma_N^2 \cdot (1-q^2)^{N-1}.$          (30)

### 2.3.4    Acceptance ratios

For each type of proposal to move along the chain from $\mathbf{m} \to \mathbf{m}'$ we derive the acceptance probability, $\alpha$, using Eq. (11), and we address each proposal type in turn. Only if the right hand side of Eq. (11) is greater than or equal to some random number drawn from a uniform distribution between 0 and 1 will a proposal be accepted. To avoid potential calculation problems with 10   large exponentials, we take the log of both sides of Eq. (11).

Since the proposal ratio for an emissions update is 1, the acceptance term is the conventional, fixed-dimension MCMC acceptance. The acceptance term can be formed from Eq. (19) and Eq. (23). Assuming a lognormal prior PDF of emissions the acceptance term for a change to the emissions value of basis function $x_i$ is of the form:

$$
\alpha_{x-update} = \min \left[ 1, \exp \left( \frac{-(\ln x_i' - \mu)^2}{2\sigma_x^2} + \frac{(\ln x_i - \mu)^2}{2\sigma_x^2} \right) \cdot \exp - \left( \frac{\Phi(\mathbf{m}') - \Phi(\mathbf{m})}{2} \right) \right].
\tag{31}
$$

15   We consider the case of a change in two types of hyperparameter, those acting on $\mathbf{x}$ and those on $\mathbf{y}$, denoted $\boldsymbol{\theta_x}$ and $\boldsymbol{\theta_y}$ respectively. A change in $\boldsymbol{\theta_x}$ will not have any impact on the likelihood function, and so the acceptance ratio is dependent purely on the prior probabilities of the new and proposed state. If the PDF of $\boldsymbol{\theta_x}$ is uniform, then we need only consider the prior PDF ratios on $\mathbf{x}$. Given a change to both PDF parameters of a lognormal distribution on $\mathbf{x}$, the acceptance ratio will become:

20   $\alpha_{x-hyperparameter} = \min \left[ 1, \dfrac{\sigma_x}{\sigma_x'} \exp \left( \dfrac{-(\ln \mathbf{x} - \mu')^2}{2\sigma_x'^2} + \dfrac{(\ln \mathbf{x} - \mu)^2}{2\sigma_x^2} \right) \right].$        (32)

Importantly, this hyperparameter is not directly informed by the data, and so the independence of the form of the prior from the data is not violated. However, a perturbation in $\boldsymbol{\theta_y}$, whether that be a change in the variances or correlation lengths will impact

upon the likelihood function, since a change in $\boldsymbol{\theta_y}$ will result in a change in $\mathbf{R}$. Assuming a uniform distribution for the prior PDF of the hyperparameter the acceptance term for a change in $\boldsymbol{\theta_y}$ is:

$$\alpha_{y-hyperparameter} = \min\left[1, \exp\left(\frac{-(\Phi(\mathbf{m'}) - \Phi(\mathbf{m}))}{2}\right) \cdot \frac{|\mathbf{R}|}{|\mathbf{R'}|}\right]. \tag{33}$$

As for the emissions update, the proposal ratio for a movement of a Voronoi cell is 1. In addition, there is no change in the prior distribution, since the cell that moves takes its emissions value with it, and the dimension of the model does not change. Thus both $\frac{\rho(\mathbf{x'}|k)}{\rho(\mathbf{x}|k)}$ and $\frac{\rho(\mathbf{c'}|k)}{\rho(\mathbf{c}|k)}$ equal 1. The acceptance term is dependent therefore only on the likelihood ratio:

$$\alpha_{move} = \min\left[1, \exp\left(\frac{-(\Phi(\mathbf{m'}) - \Phi(\mathbf{m}))}{2}\right)\right]. \tag{34}$$

The acceptance term for a birth takes the form of the full transdimensional acceptance given in Eq. (10). The terms involving $K$, the underlying resolution of the grid, in the prior and proposal distributions cancel each other out in the final acceptance term. In practice, this means that one does not have to define the nuclei locations as being restricted to the locations of the underlying grid, and they can in fact take any position within the inversion domain. However, since it makes little sense to solve at a resolution finer than the native resolution of the sensitivity maps generated by the CTM, in this work we continued to restrict the nuclei locations to the centre points of the underlying grid. The full acceptance term for a birth, assuming a lognormal prior emissions PDF is:

$$\alpha_{birth} = \min\left[1, \frac{\sigma_{bd}}{\sigma_x x'_{k+1}} \cdot \exp\left(\frac{(x'_{k+1} - x_i)^2}{2\sigma_{bd}^2}\right) \cdot \exp\left(\frac{-(\ln(x'_{k+1}) - \mu)^2}{2\sigma_x^2}\right) \cdot \exp\left(-\frac{\Phi(\mathbf{m'}) - \Phi(\mathbf{m})}{2}\right)\right]. \tag{35}$$

Similar to the birth proposal, the death proposal takes the full transdimensional form. Assuming a lognormal prior PDF for emissions the acceptance term is:

$$\alpha_{death} = \min\left[1, \frac{x'_k \sigma_x}{\sigma_{bd}} \cdot \exp\left(-\frac{(x'_j - x_k)^2}{2\sigma_{bd}^2}\right) \cdot \exp\left(\frac{-(\ln(x_k) - \mu)^2}{2\sigma_x^2}\right) \cdot \exp\left(-\frac{\Phi(\mathbf{m'}) - \Phi(\mathbf{m})}{2}\right)\right]. \tag{36}$$

At each iteration, movement along the chain is governed by the acceptance ratios given in Eq. (31), Eq. (34), Eq. (35) and Eq. (36). If the proposal is accepted, the basis function model moves to this new state, and the next proposal is made based on this new state. If the proposal is rejected, then the basis function model remains unchanged, and a new proposal is made based on the same state. In this manner, one is able to explore the space of the posterior PDF, $\rho(\mathbf{m}, \boldsymbol{\theta}, k|\mathbf{y})$. A pseudo-code example that summarises the form of the reversible jump algorithm as set out above is given in Algorithm 1.

The chain must be run for a sufficient number of iterations in order for convergence of the posterior distribution to occur. The convergence refers to the stability of the distribution across the sampled iterations of the Markov chain. In the fixed-dimension case well established convergence assessments exist, by examining convergence of each element of the parameters vector $\mathbf{x}$. However, this is not as straightforward in the transdimensional case, since an element of $\mathbf{x}$ will refer to a different region

of the domain at different points along the chain, or may not even exist. However, the convergence of the fixed dimension hyperparameters, or the fit of the predicted data values could also be natural candidates for convergence assessment (Green, 2003). Alternatively, another useful indicator of convergence may be to examine the emission values of the underlying fine grid (Bodin and Sambridge, 2009). A cursory examination of the trace of an underlying grid cell along the chain, can often be

enough to adjudge whether convergence has occurred or not. More formally convergence can be assessed using a metric such as Geweke's diagnostic (Geweke, 1992).

### 2.3.5    The posterior distribution

In order to achieve a stationary posterior distribution for the parameters, the number of iterations for which the chain must be run is large, of the order of $10^5 - 10^6$. Since the state of the basis functions may not progress from one iteration to the next,

one can perform a thinning of the full chain, storing only a subset, such as every $100^{th}$ iteration. Each stored iteration will have a different arrangement of regions, and parameter values. The coarse regions of each iteration can be mapped back onto the underlying fine resolution of the native grid. Each discrete point on the Markov chain is unlikely to be much more meaningful than any other, since each one may contain a relatively coarse partitioning of the spatial domain. Instead, the solution is the full posterior PDF (i.e. all stored iterations). From this PDF we can extract quantities of interest, such as the mean or median

and uncertainty range. Each underlying native grid cell will belong to many different regions during the course of movement along the chain. The mean of the posterior PDF for each underlying grid cell provides a naturally smoothed solution (i.e. at the resolution of the underlying finite grid), without the need to specify any specific correlation coefficients between grid boxes.

### 3    The Chemical Transport Model

A key component of the likelihood function is $\mathbf{H}$, which gives the sensitivity of mole fractions at an observation site to a change

in emissions from a finite regular grid. In order to calculate this sensitivity matrix we use the UK Met Office's Numerical Atmospheric dispersion Modelling Environment (NAME, Jones et al., 2007; Manning et al., 2011). NAME is a Lagrangian particle dispersion model, which tracks model particles backwards in time from a release point, and calculates their interaction with the surface over a given number of preceding days.

     In the pseudo-data and real data examples discussed in section 4 and section 5, NAME was run by releasing 20,000 model

particles per hour, in a vertical column of $\pm 20$ m, surrounding the location of the measurement inlet heights. The transport of the particles was driven by meteorology from the UK Met Office's Unified Model (UM). The sensitivity of the measurements to the flux from each grid cell was output by calculating the integrated residence time of the particles in a layer adjacent to the surface (0 to 40 m agl).

     Particles were tracked backwards for 30 days in a large regional domain with bounds of $(-98°E, +40°E)$ longitude, and

$(10°N, 80°N)$ latitude. The domain size was 391x293 grid cells, with a resolution of $0.352°$ longitude and $0.234°$ latitude. This output resolution was used as the maximum resolution underlying finite grid in the transdimensional inversion. The 30 day period of tracking was chosen to be sufficiently long such that the vast majority of the particles would exit the domain within

the back-trajectory period. When a particle left the modelling domain, the exit location was stored in longitude and height on the N and S sides, and latitude and height on the E and W sides. This information was then used to predict the "baseline" contribution, which is the modelled mole fraction that could not be explained by emissions from within the NAME domain. Further details of this calculation are given in the supplement. The output of the NAME model provided an estimate of $\mathbf{H}$ for each time step, which was multiplied by an emissions field to create a time series of modelled mole fractions.

## 4   Pseudo-data example

In order to demonstrate the utility of the transdimensional inversion framework, we applied it to a pseudo-data example, where the true emissions field was known. An emissions field of anthropogenic methane was taken from the Emissions Database for Global Atmospheric Research (EDGAR, EC-JRC/PBL, 2011). This time-independent field was regridded from the native resolution of 0.1x0.1 to the coarser NAME output resolution of 0.234x0.352, for an inversion domain which covered a section of North-West Europe (NWEU). This relatively small domain contained $56 \times 48$ grid cells at the native NAME output resolution. Taking the EDGAR field as the prior (shown in Fig. 2(a)), the emissions field was scaled, such that emissions from certain regions were some multiple of the EDGAR total for that country, as shown in Fig. 2(b). This chequerboard pattern shows regions where the true emissions were greater than the prior (red) and less than the prior (blue), with hard boundaries between them. This scaled chequerboard emissions field was then taken to be the true emissions field, which the inversion should attempt to retrieve. This true emissions field was multiplied by the NAME footprints at each time step to create a time series of pseudo-observations. Pseudo-observations were created at 4 sites across the UK and Ireland, that make up the UK DECC network (Ganesan et al., 2015), using 6-hourly averaged NAME sensitivities from a two-month period May-June 2014. This gave a total of 942 pseudo observations from the four sites, the locations of which are shown in Fig. 2(b). Random noise equating to a standard deviation of $\pm 5$ ppb was added to this pseudo-data, to simulate model-measurement errors.

The pseudo-data inversion was first performed in the traditional Metropolis-Hastings sense, using a fixed grid with random arrangements of 4,8,16,32,64,128 and 256 Voronoi cells as the basis functions. For each fixed number of cells, 500 different random arrangements of the cells were used, in 500 separate inversions. Within each cell the distribution of the prior emissions field was fixed, such that perturbations to the value of a cell represented a scaling of the underlying emissions distribution within that cell. The initial a priori scaling was 1 throughout the domain, compared to the true chequerboard pattern which had values of 1.5 and 0.5 in the regions of high and low scaling respectively. The inversion was performed in a non-hierarchical sense, so that the model-measurement uncertainty was fixed to be the true value of 5 ppb. The prior emissions uncertainty was fixed at 100 % of the initial value.

For each experiment the root mean square error (RMSE) of the posterior mean modelled mole fractions minus the true observations (without added data noise), was taken as a measure of the fit to the true data. The mean and standard deviation of these RMSE values across the 500 inversions are shown in Fig. 3 by the blue line and shading. As one might expect, increasing the number of regions led to a better fit to the true data, due to the greater number of degrees of freedom in the parameters.

Since the RMSE represents the difference between the posterior and true mole fractions, rather than the posterior and noisy mole factions, the RMSE is able to be less than the data noise of 5 ppb.

An additional experiment also used a fixed set of basis functions, but in such a configuration that had been designed to be higher resolution close to the measurement sites and follow national boundaries. This grid was based on the set up of Ganesan
et al. (2015) with a total of 94 basis functions in this small inversion domain. The RMSE of the posterior modelled mole fractions, shown as a yellow triangle in Fig. 3, was slightly lower than the mean of random grids of the equivalent number of regions. The RMSE for this subjectively determined grid of 2.0 ppb is similar to the expected value of a random arrangement of 256 Voronoi cells. The relative performance of this expert judgement grid can be explained in part by the fact that the boundaries between regions were specified by horizontal and vertical lines, like the true chequerboard pattern. However, those
boundaries are not always in the right place to match the true chequerboard, as shown in Fig. 4(a), meaning that there is a limit to the improvement in fit to the data.

Although the overall pattern of reds and blues is discernible in Fig. 4(a), the scaling map is characterized by a number of more extreme high or low values, depicted by darker reds and blues, which do not exist in the true field, as well as the incorrect specification of the boundaries. Of course, with real data the true patterns are not something we can know a priori. As was the
case in this example, a subjective choice of basis functions can preclude the recreation of the true emissions field, no matter how much information the data contains.

Finally the inversion was performed using the transdimensional approach, where the number and configuration of basis functions was allowed to vary. Forty regions were chosen a priori, and the bounds of the uniform prior were 5 and 500 unknowns. Each discrete point on the transdimensional Markov chain contained a relatively coarse partitioning of the spatial
domain, which may have provided an RMSE little better than a randomly chosen grid of the same number of regions. However, since the solution is the entire posterior PDF of the parameters, we can extract the mean value of the posterior distribution for each of the underlying grid cells, and use this to recreate a set of mole fractions. This naturally smoothed solution gave a significantly reduced RMSE in the data space for the mean number of regions, shown by the green circle in Fig. 3. The RMSE value of 1.0 ppb was smaller (approximately a half) than that of the subjectively determined grid, for this particular
pseudo-data example. The equivalent expected RMSE for fixed random grids of the same number of regions as the mean of transdimensional posterior distribution was around 6 ppb, showing the effective gain achieved by sampling from many different basis function configurations, rather than just one.

Figure 4(b) shows how the shape of the transdimensional posterior mean scaling field is similar to the true field in the areas which are sufficiently well seen by the data. The boundaries between areas of higher and lower emissions were resolved
almost exactly in some central areas of the domain, and the scaling magnitudes correspond better to the true values, with fewer extremes. The effects of the data noise are seen in the failure of the transdimensional solution to resolve all the hard boundaries of the chequerboard regions precisely. Furthermore, if we had more measurement sites in those areas of the map that are not well seen by the current data, we might expect the solution to provide an even better fit to the true solution.

This point is highlighted by the estimated uncertainty map which is extracted directly from the posterior PDF, shown in Fig.
4(c). This shows that the areas of highest uncertainty were generally where there was very little constraint by the data, such as

over the oceans where the emissions were comparatively negligible. Since there was little data to constrain proposals in these regions, we ended up exploring the prior PDFs to a wider extent, leading to a larger posterior uncertainty. In addition, darker features of greater uncertainty are visible in the grid cells between regions of higher and lower emissions. This is in line with what one might predict, the central parts of the chequerboard regions are well constrained, but where the boundaries lie is not as well known.

Of course, if one specifies the form of the basis functions correctly, then the model-measurement RMSE can be minimised. This is shown by the magenta square on Fig. 3, which has an RMSE value of 0.6 ppb and represents the result from an inversion with 16 basis functions in the same arrangement as the true chequerboard pattern. However, while it may be possible to achieve this in a heavily simplified pseudo-data example, the reality is that we can never know a priori the exact form the basis functions should take to minimise the posterior RMSE. In this case, we see that the naturally smoothed transdimensional solution contains a larger degree of truthfulness, when compared to either random or subjectively specified fixed grids.

The posterior distribution on the number of derived unknowns in the transdimensional solution is shown in Fig. 5. The range of the posterior distribution was well within the bounds of the uniform prior, showing the constraint that the data had on this quantity. The true number of regions in the chequerboard pattern was 16, whereas the mean of the posterior distribution was slightly larger at $29 \pm 7$ . Primarily, the Voronoi nuclei were concentrated around the areas that are best seen by the data (UK and Ireland), and there was most variability in position in those areas that were not well constrained by the data, particularly over the sea. Since changes to the configuration of regions in these poorly seen areas may have little impact on the likelihood ratio, any proposal to change the Voronoi cells in these areas should be more frequently accepted. Indeed, if there were no data at all, then we would expect merely to explore the prior distribution for the number of regions and their emission magnitudes across the whole spatial domain.

## 5 Real data inversion

The pseudo-data example shows the merits of the transdimensional approach when there exist hard boundaries between areas of over or underestimation in the prior. In reality, such clear cut scaling fields are unlikely, and so it is pertinent to observe how the inversion performs when confronted with real data. In order to achieve this, we performed an inversion using one month of $CH_4$ data from the UK DECC network (Ganesan et al., 2015). While one might expect real world trends in emissions to follow national or regional boundaries, this is by no means guaranteed, or indeed distinguishable by the data. The transdimensional approach allows us to determine the patterns in emissions at a resolution that the data allows.

### 5.1 UK methane emissions using the UK DECC network

Compared to the pseudo-data experiment above, in addition to solving for the emissions scaling factors and the number of unknowns, various hyperparameters were also considered variable, which were to be solved in the inversion. Hyperparameters describing the prior log-mean and log-standard deviation, model-measurement uncertainty and auto-correlation timescale were each described by a uniform PDF. Prior emissions were taken from the EDGAR emissions inventory (EC-JRC/PBL, 2011) for

2010. For simplicity, we ignore natural sources in this study. Ganesan et al. (2015) showed that natural sources contributed to less than 10% of the UK's emissions, and that their omission had little effect on the derived net flux.

An emissions field was estimated for March 2014 using data from the four measurement sites of the DECC network: Mace Head, Ireland; Tacolneston, England, Ridge Hill, England and Angus, Scotland, shown in Fig. 2(b). Measurements were av-
eraged from one minute frequency into 4-hourly periods, giving a total of 727 data points. A different model uncertainty parameter governed each seven-day period, consistent with a typical timescale of synoptic variability. These seven-day periods were further divided to estimate separate uncertainties for times when there was a significant degree of "local influence" on the measurement site. These local events were represented by times when the fraction of the NAME sensitivity footprint from the nine grid boxes surrounding each station was greater than some threshold. A high local fraction represents times when
the air might be particularly stagnant, and transport is influenced by sub-grid scale processes which the model cannot resolve, such as local land-sea breezes. Solving for a separate model uncertainty at these times allowed us to weight the corresponding measurements appropriately. The threshold chosen as the degree of localness was 30% of the total sensitivity of the NAME spatial domain.

The total NAME output domain was of dimension 391x293, however we restricted the domain on which the transdimensional
inversion would be carried out to a much smaller 64x52 grid. Outside of this sub-domain, the emissions distribution was assumed fixed in six separate regions, shown in Fig. 6, with the scaling of each region solved for in the inversion. To account for the mole fractions that could not be explained by emissions from the local inversion domain over the 30 day NAME back trajectory period, scalings to the mole fraction field arriving at the 4 edges of the NAME domain were also solved for in the inversion, as described in the supplement. The transdimensional inversion was performed with a uniform prior on the number
of regions defined between 5 and 800 unknowns. A burn-in period chain of 100,000 iterations was run and discarded, ahead of 500,000 iterations, with every 100th iteration along the chain being stored. Results are given as the mean of the posterior PDFs, whilst uncertainties for all parameters correspond to the 5th to 95th percentile range.

Figure 7(a) shows the scaling of the prior emissions field required to form the posterior field. The derived posterior indicates an overall decrease in UK and Ireland emissions from the prior, although there are exceptions most notably in southern England,
south Wales, Tyneside and Merseyside regions (in the northwest and northeast of England respectively). Other areas where emissions in this month were estimated to be higher than the prior are clearly seen near the French/Belgian border and Brittany. The increase around the Mace Head station may be indicative of poor resolution of local transport, or a local pollution source not included in the prior such as peatland emissions. Conversely, a large decrease from the prior is seen over the Paris area. Total UK emissions in this month were found to be 2.28 (2.04–2.52) Tg yr$^{-1}$, and Ireland 0.49 (0.39–0.60) Tg yr$^{-1}$, a decrease
from the prior of 2.80 $\mathrm{Tg yr}^{-1}$ and 0.63 $\mathrm{Tg yr}^{-1}$ respectively. This is in line with the results reported by Ganesan et al. (2015) of 2.05 (1.60–2.70) Tg yr$^{-1}$ and 0.50 (0.41–0.61) Tg yr$^{-1}$ respectively. The results of Ganesan et al. (2015) were based on the same DECC data, averaged into 2-hourly periods, using a hierarchical Bayesian approach. Differences between the methods lie in the transdimensional scheme implemented here, the method of accounting for baseline mole fractions (see Supplementary Material), and the prior fluxes. The UK and Ireland estimates were found to be stable with respect to the number of iterations

from which the posterior distribution was sampled. This shows that the burn-in period was sufficient for convergence of these national scale emission totals to occur.

The mean number of unknowns was found to be 201 (145–248), as shown in Fig. 8. This rather wide distribution can perhaps be explained by the lack of data constraint on the parts of the inversion domain that contain relatively small emissions, are far removed from the measurement sites, and hence are not well seen by the data. Nevertheless, the transdimensional inversion can give us an idea of the resolution at which the data is able to infer differences in emission patterns. On the whole it appears as if changes to the prior are made on a relatively large regional scale, which may reflect a large-scale bias in the prior. The regions of higher emissions appear to be resolved at the resolution of a few grid cells, where each grid cell corresponds to a roughly $25 \times 25 \text{ km}^2$ area.

In addition to inference on the mean of the posterior distribution, the posterior PDF of the emissions field gives us a direct estimate of the uncertainty of each grid cell. Figure 7(b) shows the uncertainty reduction for this month, defined as $1 - \left( \frac{90\%ile \text{ range posterior}}{90\%ile \text{ range prior}} \right)$. It shows how the greatest reductions were in major emissions areas which are in proximity to the measurement sites. Outside of the UK the uncertainty reduction is seen to be fairly minimal, which is entirely consistent with the decay in sensitivity with distance form the measurement sites. This further tallies with the somewhat noisy scaling patterns seen over the oceans in Fig. 7(a), where the data is unable to infer much useful information given the substantially smaller fluxes.

Inference on the various hyper-parameters of interest can inform us about the relative modelling performance at each of the measurement sites. Modelling uncertainties were found to be smallest at the Angus site, with a mean uncertainty of 8 (4–15) ppb. This is consistent with the station mainly sampling clean air, being far enough away from large, variable emission sources in Scotland. In contrast, higher modelling uncertainties were derived for the Tacolneston and Ridgehill stations, of 32 (10–73) ppb and 25(8–58) ppb respectively, consistent with both sites intercepting polluted air more frequently. The average correlation time scale, based on the prescribed exponential decay structure, was found to be 15 (7–37) hours across all four sites. No significant difference was found between the uncertainties derived for times when local influence was high and those when it was not. By contrast, Berchet et al. (2013) reported $CH_4$ observation uncertainties that were on average 23–31% smaller during the day than at night for a number of sites across Europe using three different hyperparameter optimization schemes. Errors in boundary layer modelling are likely to be greater at night, although these may be more systematic than random. A better understanding of modelling uncertainties, and how they can be accounted for in the hierarchical framework would be necessary to include this potential bias.

## 6  Discussion and further development

The inversion above took around 90 minutes to run 600,000 iterations, on a single processor, although there were two primary time consuming steps that affected the computation time. The first was calculating the inverse of the model-measurement covariance matrix. In the above example, there were around 750 observations, so that the inverse had to be calculated on a 750x750 matrix, an expensive step each time a temporal correlation parameter was changed. To avoid this, the measurements

were assumed to be spatially uncorrelated, so that the covariance matrix was of block diagonal form. This assumption was made following Ganesan et al. (2015), who found a mean correlation length scale of around 100 km for $CH_4$ using the DECC network. This distance is significantly less than the minimum distance between any of the stations of 250 km. To simplify things further, if the data is assumed to be evenly spaced in time, then a simple analytical solution exists for the inverse

(Malinverno and Briggs, 2004), and inverting each block need not be so computationally expensive. However, if large gaps in the data exist, (due to instrument downtime, flagged observations etc.), then this assumption would not be appropriate. In such cases it may be necessary to reduce the dimension of the data space further or to assume a fixed correlation timescale to make the inversion feasible.

The other rate-limiting step is the recalculation of the Voronoi cells and the associated sensitivity of each one, every time

a birth, death or move is proposed. In practice, this need not be recalculated for all Voronoi cells, only those that change in moving from the current state to the proposed state. However, this can still be a cumbersome calculation. The use of Voronoi cells present a simple, albeit rather crude approach to the partitioning of the inversion domain, and it is our hope to extend this method to other forms of basis functions in the future (e.g. Hawkins and Sambridge, 2015). Furthermore, although issues with low acceptance ratios can often occur in transdimensional inversions (Bodin and Sambridge, 2009), this was a problem that

was not immediately apparent in our inversion. However, this could have been due in part to there being large regions of the inversion domain that had little or no constraint by the data. High acceptance ratios in these areas could mask low acceptance ratios in better constrained parts of the domain. Such a case points to inefficiencies in our inversion framework, that could be improved by an alternative definition of the basis functions.

Although the examples above were run using a single chain on a single processor, the opportunities for running multiple

chains in parallel should be readily apparent. The implementation of several independent chains run in parallel on multiple processors could allow for significantly fewer iterations being required for each chain. Indeed, a further development is to have several chains able to communicate with each other throughout the inversion, where each chain is tempered by a given parameter. This technique of parallel tempering has the potential to allow vast improvements in efficiency when compared to the conventional Metropolis-Hastings algorithm, especially for multi-modal PDFs (Sambridge, 2014).

In this work, we intentionally chose to focus only on the 2-D spatial aggregation of the fluxes and ignored the assumptions made in aggregation of the temporal dimension due, primarily, to concerns about the computational demands of extending this particular implementation to 3-D (Piana Agostinetti et al., 2015). However, there is no inherent reason that the transdimensional approach could not be further extended to the 3-D problem. Such an extension would inevitably incur higher computational expense, particularly with the frequent need to recalculate 3-D Voronoi cells. It may be possible to ameliorate these demands

by prescribing an alternative form of basis function such as a tree structure similar to (Bocquet et al., 2011), which may be both faster to calculate and more efficient at exploring the 3-D parameter space (e.g. Hawkins and Sambridge, 2015).

This inversion framework is inherently suited to cases where one does not have to continuously recalculate the native resolution sensitivity matrix, $\mathbf{H}$, as is the case here for the Lagrangian model output. As such, we anticipate this method may be restricted to use with CTMs that are capable of calculating underlying sensitivities at a fine grid scale. It is further apparent

that the useful information that can be determined from the data decays quickly with distance from the measurement sites.

## 7 Conclusions

We have demonstrated how reversible-jump Markov chain Monte Carlo can be applied to inverse modelling of trace gas emission fields. In allowing the number of unknowns itself to be an unknown, the method attempts to avoid some of the assumptions that have had to be made in atmospheric inverse modelling. Furthermore, the uncertainty surrounding the choice of number and shape of unknowns, propagates through to the posterior distribution. We have shown how, through making a reduced set of assumptions about the shape and number of our basis functions, this transdimensional approach can lead to a better fit to the data, and an improved representation of a true emissions field, when compared to a random or subjective basis function definition. Combined with a hierarchical framework, the method set out here is focused on using the data to as great an extent as possible to guide our solution. Emissions derived using the transdimensional hierarchical framework, from the UK and Ireland during March 2014, were found to be consistent with previous work. The framework provides an alternative approach to using a single partitioning of basis functions when performing dimension reduction.

## 8 Code and data availability

NAME is a UK Met Office model available for external research use under license. Information on obtaining a license can be obtained by contacting the Met Office directly. The reversible jump MCMC Fortran code can be obtained upon request. Data from the UK DECC network is available for download from the EBAS database: http://ebas.nilu.no/.

*Acknowledgements.* We thank Simon O'Doherty, Aoife Grant, Dickon Young and all the site operators of the DECC network for their tireless work and dedication to providing high quality and reliable data. Mark Lunt is funded under a studentship from the UK Natural Environment Research Council (NERC). Matt Rigby is funded by a NERC advanced research fellowship NE/I021365/1. Anita Ganesan is funded under a NERC independent fellowship NE/L010992/1.

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

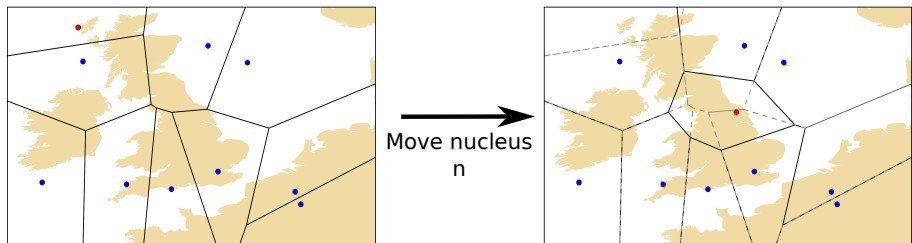

**Figure 1.** A spatial domain partitioned into 10 Voronoi cells. Moving one nucleus to a new position in the domain changes the boundaries of the neighbouring cells.

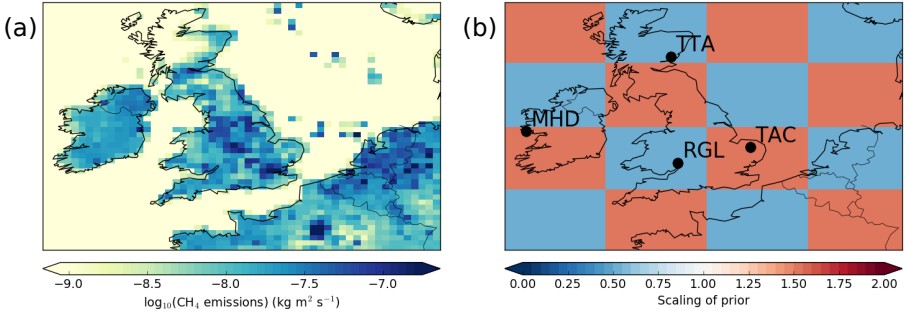

**Figure 2.** a) The underlying EDGAR methane emissions mapped onto the NAME output resolution, used as the prior in our inversions. Note the log scale. b) The chequerboard scaling pattern applied to the EDGAR emissions to create the true emissions field. The location of the DECC sites used are shown by the black dots showing (from North to South): Angus, Scotland (TTA), Mace Head, Ireland (MHD), Tacolneston, England (TAC) and Ridgehill, England (RGL).

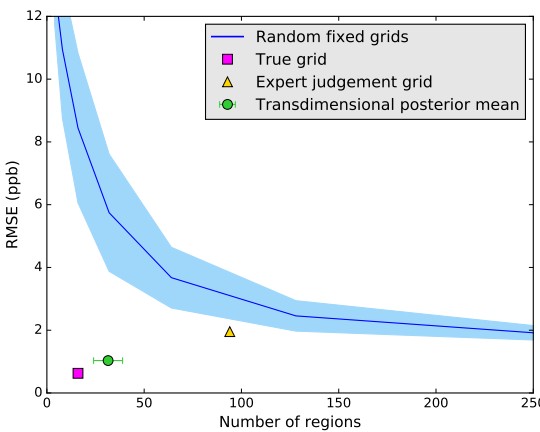

**Figure 3.** RMSE in the data space as a function of number of unknowns, given as $(\mathbf{y_{posterior}} - \mathbf{y_{true}})$. The mean of the transdimensional posterior distribution provides a significantly improved fit to the data.

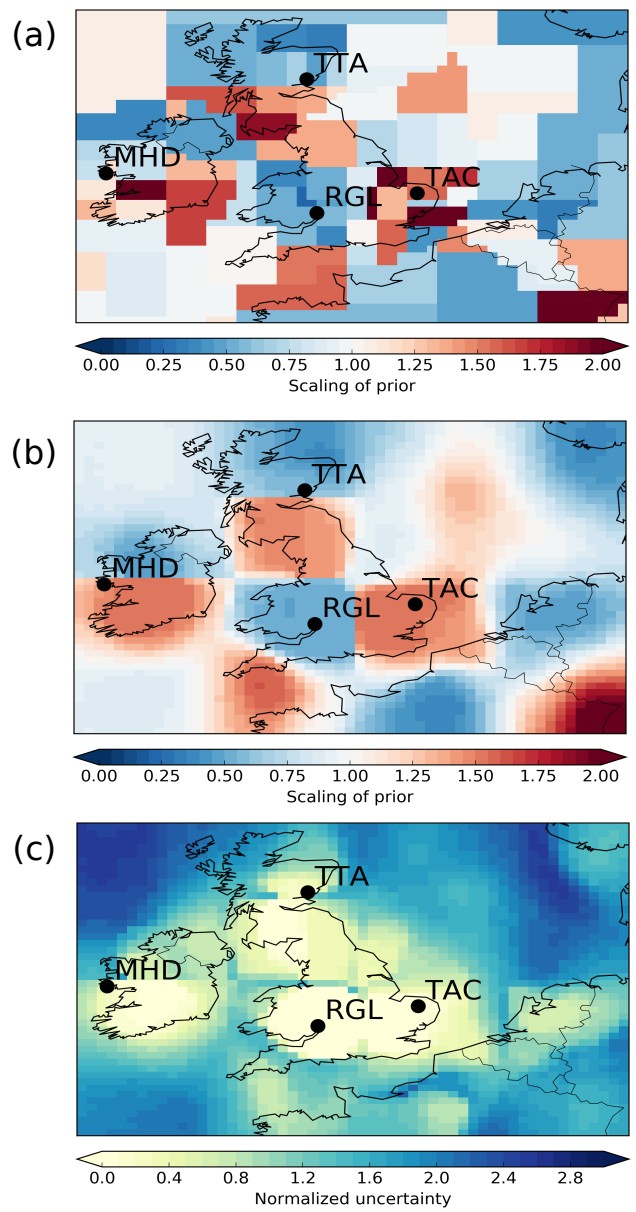

**Figure 4.** Maps of the posterior scaling of the prior for the subjectively optimised fixed grid (a) and the mean of the transdimensional inversion posterior (b). The uncertainty of the transdimensional posterior is shown in (c), defined as the normalized 90 %ile range.

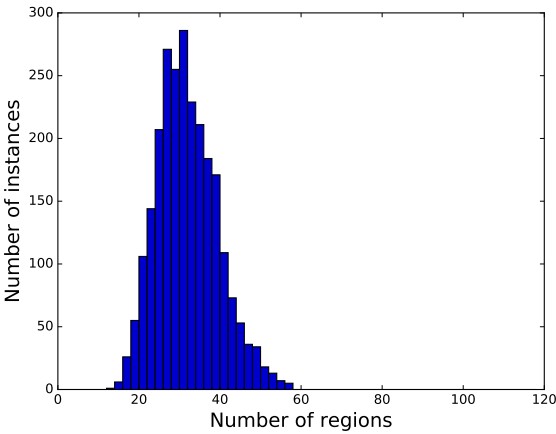

**Figure 5.** Posterior distribution of the number of unknowns in the pseudo-data experiment.

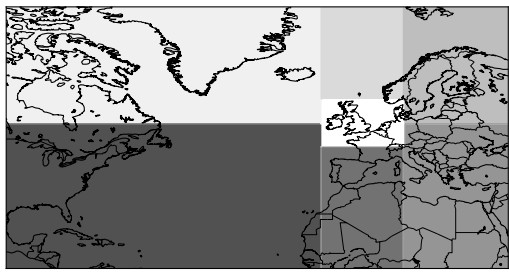

**Figure 6.** Schematic of the arrangement of the full NAME computational domain, divided into six fixed regions and the sub-domain in which the transdimensional inference was performed, shown by the different shadings.

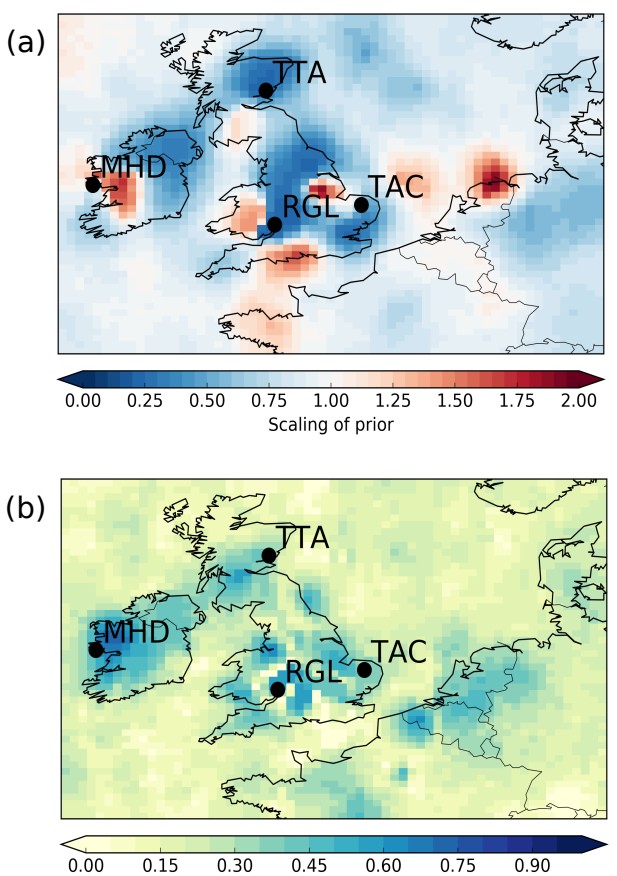

**Figure 7.** (a)Ratio of the posterior to the prior emissions field. (b) Map of the posterior uncertainty reduction defined as $1 - \left( \frac{90\%ile \text{ range posterior}}{90\%ile \text{ range prior}} \right)$. The 90%ile range represents the 5th to 95th percentile range.

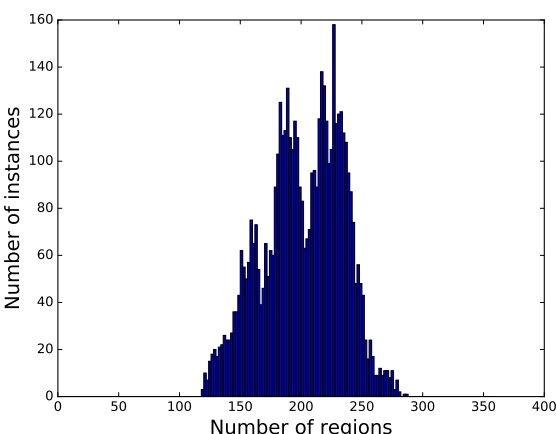

**Figure 8.** Posterior distribution of the number of unknowns in the real data inversion.

**Algorithm 1** Reversible-jump pseudo code

---

**for** i = 1,n **do**

    $r \leftarrow i \bmod 5$

    **if** $r = 0$ **then**

        $\mathbf{x'_i} = \mathbf{x_i} + N(0, \sigma_x)$  {Emissions update}

    **else if** r=1 **then**

        $\boldsymbol{\theta'_i} = \boldsymbol{\theta_i} + N(0, \sigma_{theta})$  {Hyperparameter update}

    **else if** r=2 **then**

        $k' = k + 1$ {Birth}

        Form new Voronoi cells

    **else if** r=3 **then**

        $k' = k - 1$ {Death}

        Form new Voronoi cells

    **else if** r=4 **then**

        $\mathbf{c'_i} = \mathbf{c_i} + N(0, \sigma_{move})$  {Move}

        Form new Voronoi cells

    **end if**

    $\alpha \leftarrow (\mathbf{x'}, \mathbf{c'}, \boldsymbol{\theta'}, k')$  {Calculate acceptance ratio}

    **if** $\ln(\alpha) \geq \ln(U(0,1))$ **then**

        $(\mathbf{x}, \mathbf{c}, \boldsymbol{\theta}, k) = (\mathbf{x'}, \mathbf{c'}, \boldsymbol{\theta'}, k')$ {Accept}

    **end if**

    $q \leftarrow i \bmod 100$

    **if** q=0 **then**

        Store $(\mathbf{x}, \mathbf{c}, \boldsymbol{\theta}, k)$ {Store every 100th iteration}

    **end if**

**end for**

---