# Peer review of "Estimation of trace gas fluxes with objectively determined basis functions using reversible jump Markov Chain Monte Carlo"

_Geoscientific Model Development, 2016_

## Referee Comment (RC1) · Anonymous Referee #1 · 22 Jun 2016

Lunt and coauthors have written, and well written, an interesting paper about dimension reduction for atmospheric inverse modelling. The introduction of the dimension itself as an unknown directly in the Bayesian emission estimation is particularly original. I can only recommend it for publication, provided a few points are addressed:

- The authors have restricted the dimensionality problem to the 2D space, and seem to have forgotten that the problem has a temporal dimension as well. This question should be addressed somehow right from the start of the paper and in the pseudo-data example.

- P. 2, l. 10: the correlation is on emission errors, not on emissions.

- P. 2, l. 15: "that do not exist in the true field" actually applies to any flux estimate, since it remains uncertain.

- P. 4, l. 22: to be fair, the authors should also cite earlier publications like Michalak et al. (2005, doi : 10.1029/2005JD005970), Berchet et al. (2013, doi:10.5194/acp-13-7115-2013) or Wu et al. (2013, doi :10.3402/tellusb.v65i0.20894).

- P. 5, l. 25: "its".

- P. 6, l. 7: "unintelligent" sounds harsh and a softer word would be more appropriate.

- P. 7, l. 7: the authors should also refer to earlier studies.

- P. 7, l. 10: the validity of this assumption should be discussed. At first glance, it looks poor. For instance a large dependency in the case of natural $CO_2$ fluxes over land was shown by Chevallier et al. (2012, doi:10.1029/2010GB003974, their Fig. 5). More generally for instance, it is very likely that hyperparameters are not the same at city-scale and at country-scale.

- P. 9, l. 14: how can the prior location and emissions variables be independent of each other?

- P. 13, l. 7 and l. 9: why is there a notion of convergence (l. 9; like if we were looking for just the most-likely state) while the algorithm explores the space of the posterior pdf (l. 7 and 24)?

- P. 13, l. 19: what does "typically" mean here?

- P. 13, l. 27: why should the solution of the problem (independent of the resolution method) be smooth? In other words, is it an advantage or an inconvenient to generate a smooth solution?

- P. 15, l. 10: is "twice as small" significant here?

- P. 17, l. 8: how stable is the estimate with respect to the number of iterations?

- P. 18: the first paragraph on the page reminds of the discussion by Berchet et al. (2013, doi:10.5194/acp-13-7115-2013, their sections 3.1, 3.2 and 3.3) on the same topic. This may be acknowledged.

- P. 18, l. 11: "To avoid this, the..."

- P. 19, l. 6: the sentence is too trivial to be the last one.

---

## Referee Comment (RC2) · Anonymous Referee #2 · 22 Jun 2016

**1   Main comments**

This paper represents a very interesting description and application on synthetic and real data of tracer, of the most recent techniques in transdimensional inverse modelling. In that respect, this paper is very valuable and I believe that it should ultimately be published. However, the manuscript is not without flaws that should be corrected before the paper becomes acceptable for publication. The most important flaws to be corrected are

1. From time to time, a crucial lack of details.

2. The bibliographical account on hyperparameter estimation is very misleading. As opposed to what is implied in the manuscript, this subject has been addressed in tracer inversion and greenhouse gas inversion for more than 10 years now. I have some knowledge on all the techniques that are addressed by the authors, and I can tell that this poor bibliographical account clashes with the rather good account on the other mathematical aspects.

3. Several passages of the manuscript are odd and difficult to understand. This tells me that the manuscript has not been polished enough yet, although it is already quite enjoyable.

**2   Minor points or comments related to the main points**

1. Page 1, line 9, "it allows the uncertainty in our choice of aggregation to be carried through to the solution" is too vague for the abstract. Please clarify or postpone this statement.

2. Page 1, Eq.(1): It is common practice to insert equations within the flow of the article and use punctuation marks at the end of equations (or not if embedded within a sentence).

3. Page 2, line 16: "...or the solution being overly influenced by an incorrect prior, giving the so-called smoothing error": This statement is partially misleading to me. The fact that there are more degrees of freedom is not an issue *per se*. This has been shown in Bocquet et al. (2011): in theory the more resolved the grid, the better the inversion. It is in addition the fact that the prior could be incorrect that may lead to the smoothing error. If the incorrectness in the prior is low, then such a balance might be pointless. In that case, dimensional reduction is essentially

only important for computation issues (which is critical) as pointed out by Bocquet et al. (2011) and ultimately confirmed in Turner and Jacob (2015).

4. Page 2, line 28: "Various studies..." Obviously there has been only a few studies so far. Please mitigate your statement.

5. Pages 2, line 33-35: "Although a parameter dimension was successfully identified which minimised the total error, ultimately the choice of model to use was as much influenced by computational efficiency, as it was by this combination of aggregation and smoothing error": Yes, just as predicted in Bocquet et al. (2011). This could be mentioned.

6. Page 3, line 2-3: "Therefore, the uncertainties in step one do not necessarily propagate through to step two." All of he objective criteria in Bocquet et al. (2011) depend on the observation network. One of the criterion in Bocquet et al. (2011) actually depend on the data itself (section 4.1.3 and illustrated on Fig. 5 of the same paper).

7. Page 4, lines 20-30: This paragraph gives a wrong and totally biased picture of the literature on the hyperparameter estimation as used in tracer/greenhouse gas inversions, not to mention geophysical data assimilation and in particular meteorology. There are dozens of papers on the subject before the contributions of Ganesan et al. Only focusing on tracer inversions, one of the very first use for the inversion of the Chernobyl source term is in Davoine and Bocquet (2007) which has been extended to non-Gaussian inversion problems in the Fukushima case (Winiarek et al., 2012). But there really are dozens of papers on the subject. Two reviews on the matter are Michalak et al. (2005) and Wu et al. (2013). It would be fair to mention those papers before mentioning Ganesan et al.

8. Page 5, whole section 2.1: this subsection 2.1 is totally off in the flow of the paper, especially starting from line 10. I do not understand why the past tense is used.

I do not understand which numerical experiment you are referring to? This must be re-written or postponed in the manuscript.

9. Page 5, section 2.1: The use of a Lagrangian model such as NAME adds further interesting issues that were discussed in Koohkan et al. (2012). There is an additional uncertainty due the number of particles, especially when just a few of them fall into grid-cells. This issue could conflict with the transdimensional approach.

10. Page 5, line 25: "Furthermore, each grid cell within each aggregated region has an enforced correlation to it's neighbours": Why?

11. Page 5, line 25: "it's" $\longrightarrow$ "its".

12. Page 6, lines 19-24: "In this work, we approximate the form of the Voronoi cells, by restricting them to those points on the underlying finite grid which are closest to their respective nuclei. As such, the region edges are not exactly equidistant between nuclei, but this approach was taken since the exact form of the Voronoi cells is unimportant, and each underlying grid cell belongs to only one nuclei, making computation very simple." This passage is very unclear to me. Please give more details.

13. Page 6, line 7: "as shown by Ganesan et al. (2014)." You are pushing the envelope too far here. This was well known and shown a long time before Ganesan et al. (2014). Please remove this statement which is biased.

14. Page 6, line 10: "Where the assumption has been made that $\theta$ is independent of $k$": This is a question that has puzzled me for a long time. It may very well be that this independence is plain wrong and that it has a strong impact on the resulting inversions. At the very least, you should discuss that assumption.

15. Page 6, lines 10-20: "Whereas the hierarchical framework alone can be solved through conventional Markov Chain - Monte Carlo (MCMC) methods (Ganesan et al., 2014), since the dimension of m is variable in the transdimensional case, it must be solved by a different, though strongly related, approach." The sentence is unclear. Please rephrase.

16. Page 8, line 15, Eq.(11): $n \longrightarrow \mathbf{n}$.

17. Page 9, line 2: hyperparameter $\longrightarrow$ hyperparameter.

18. Page 9, line 3: "The other three proposals" $\longrightarrow$ "The other three proposal ratios".

19. Page 9, lines 4-5: "In effect, this means a change in the sensitivity matrix, $\mathbf{H}$, that maps the relationship between emissions and observations.": Please be more much precise as to what it implies for $\mathbf{H}$.

20. Page 9, line 9, "there are an unknown number of unknowns" $\longrightarrow$ "there is an unknown number of unknowns"?

21. Page 9, line 10, "be decomposed to two separate terms" $\longrightarrow$ "be decomposed into two separate terms".

22. Page 9, lines 10-11: "since the prior location and emissions variables are independent of each other" is not obvious to me. Can you please elaborate?

23. Page 9, lines 19: "so they may be located anywhere" $\longrightarrow$ "so they may be located a priori anywhere".

24. Page 9, line 20: "can be located on the finite underlying grid": that is an imprecise statement. Could you please be more specific?

25. Page 9, Eq.(16): Even using an – as much as possible – uninformative prior may slightly influence the number of nuclei. Can you elaborate on that?

[Figure]

26. Page 10, Eq.(18): Shouldn't $x$ be bold?

27. Page 11, line 17: "and the other a correlation length between measurements, $\tau$.": That is why I have reservation on the fact that all of the hyperparameters are independent from $k$.

28. Page 11, line 18: "it's" $\longrightarrow$ "its".

29. Page 11, line 18-21: Please elaborate. The statements are too concise.

30. Page 12, line 19: "it's" $\longrightarrow$ "its".

31. Page 12, line 24-27: What is $|\mathbf{J}|$ here?

32. Page 12, line 25: "In practice, this means that one does not have to define the nuclei locations as being restricted to the locations of the 25 underlying grid, and they can in fact take any position within the inversion domain." Okay, but what did you do in this study?

33. Page 13, line 9: "in order for convergence to occur" $\longrightarrow$ "in order for the convergence to occur".

34. Page 14, line 1: Can you describe the temporal dimension of the emissions. For instance, are they modulated in time?

35. Page 14, line 10: How many observations do you use? How long is the time frame?

36. Page 14, line 16: Which first guess (mean prior) did you choose? In general the reader is missing quite a few details to fully understand the experiment. Please give more information.

37. Page 16, line 30-32: What would happen without this filter?

38. Page 17, line 1-4: Please provide a figure.

**References**

Bocquet, M., Wu, L., and Chevallier, F.: Bayesian design of control space for optimal assimilation of observations. I: Consistent multiscale formalism, Q. J. R. Meteorol. Soc., 137, 1340–1356, doi:10.1002/qj.837, 2011.

Davoine, X. and Bocquet, M.: Inverse modelling-based reconstruction of the Chernobyl source term available for long-range transport, Atmos. Chem. Phys., 7, 1549–1564, doi:10.5194/acp-7-1549-2007, 2007.

Koohkan, M. R., Bocquet, M., Wu, L., and Krysta, M.: Potential of the International Monitoring System radionuclide network for inverse modelling, Atmos. Env., 54, 557–567, doi:10.1016/j.atmosenv.2012.02.044, 2012.

Michalak, A. M., Hirsch, A., Bruhwiler, L., Gurney, K. R., Peters, W., and Tans, P. P.: Maximum likelihood estimation of covariance parameters for Bayesian atmospheric trace gas surface flux inversions, J. Geophys. Res., 110, D24 107, doi:10.1029/2005JD005970, 2005.

Turner, A. J. and Jacob, D. J.: Balancing aggregation and smoothing errors in inverse models, Atmos. Chem. Phys., 15, 7039–7048, doi:10.5194/acp-15-7039-2015, 2015.

Winiarek, V., Bocquet, M., Saunier, O., and Mathieu, A.: Estimation of Errors in the Inverse Modeling of Accidental Release of Atmospheric Pollutant: Application to the Reconstruction of the Cesium-137 and Iodine-131 Source Terms from the Fukushima Daiichi Power Plant, J. Geophys. Res., 117, D05 122, doi:10.1029/2011JD016932, 2012.

Wu, L., Bocquet, M., Chevallier, F., Lauvaux, T., and Davis, K.: Hyperparameter Estimation for Uncertainty Quantification in Mesoscale Carbon Dioxide Inversions, Tellus B, 65, 20 894, doi:10.3402/tellusb.v65i0.20894, 2013.

---

## Author Comment (AC1) · 20 Jul 2016

We thank the anonymous reviewer for their comments on the manuscript. We have replied to each of the specific comments in turn below, with the author's response following each comment. Page and line numbers given in the author response refer to the marked up version of the manuscript, provided as a supplement to this comment.

1. The authors have restricted the dimensionality problem to the 2D space, and seem to have forgotten that the problem has a temporal dimension as well. This question should be addressed somehow right from the start of the paper and in the pseudo-data example.

[Figure]

Author's response: We agree with the reviewer that the aggregation of the temporal dimension is an important component of inverse problems, and indeed, it is something we considered carefully. However, in the manuscript we intentionally focused only on the spatial domain for the following reasons:

- There are a limited number of examples of three-dimensional transdimensional inversion in the geoscientific literature, although to our knowledge none of these involve a temporal dimension explicitly. However, one study (Piana Agostinetti et al. 2015) that used three dimensional Voronoi cells reported a computation time for the reversible jump algorithm of approximately one month, given 9700 data points and 10ˆ6 iterations of the Markov chain. Therefore, were we to explore the aggregation across three dimensions we would anticipate that a vastly more efficient procedure might be required, perhaps involving an alternative approach to Voronoi cells (e.g. Hawkins and Sambridge 2015). This would be a substantial undertaking which we believe could form an entirely new work of itself.

- The aggregation of the spatial basis functions into Voronoi cells relies on calculating the Euclidian distance between each grid cell and Voronoi nucleus. However, if this was extended to a space-time domain, it is not immediately obvious how one would calculate equivalent "distances" in space and time. One solution might be to normalise these distances, thus allowing the same Voronoi tessellation to be used across three dimensions. However, we decided that such an extension would lead to further complication and again, would be better tackled in a future paper.

In light of this comment we have made the following additions to the main text, which highlights that we have focused only on the spatial part of the problem, and we have included a discussion of how the reversible jump algorithm could be applied to the temporal aggregation of emissions in the final section.

Page 1, Line 4: "Here, we present an objective method for reducing the spatial dimension of the parameters space. . . "

Page 2, Line 10: "In addition to the spatial partitioning, some form of temporal aggregation must also be performed, over which the parameters are assumed constant. Each basis function then represents some 3-D aggregation of the underlying fluxes. In this work we choose to focus only on the 2D spatial component of emissions, making the assumption that the fluxes are constant over a fixed period of time."

Page 23, Line 11: "In this work, we intentionally chose to focus only on the 2-D spatial aggregation of the fluxes and ignored the temporal dimension in this work due, primarily, to concerns about the computational demands of extending this particular implementation to 3-D (Piana Agostinetti et al. 2015). However, there is no inherent reason that the transdimensional approach could not be further extended to the 3-D problem. Such an extension would inevitably incur higher computational expense, particularly with the frequent need to recalculate 3-D Voronoi cells. It may be possible to ameliorate these demands by prescribing an alternative form of basis function such as a tree structure similar to Bocquet et al. (2011), which may be both faster to calculate and more efficient at exploring the 3-D parameter space (e.g. Hawkins and Sambridge 2015)."

2. P. 2, l. 10: the correlation is on emission errors, not on emissions.

Author's response: We thank the reviewer for pointing out this error and have corrected this in the text.

3. P. 2, l. 15: "that do not exist in the true field" actually applies to any flux estimate, since it remains uncertain.

Author's response: We agree with this point and for the removal of doubt have removed this comment.

4. P. 4, l. 22: to be fair, the authors should also cite earlier publications like Michalak et al. (2005, doi : 10.1029/2005JD005970), Berchet et al. (2013, doi:10.5194/acp-13-7115-2013) or Wu et al. (2013, doi :10.3402/tellusb.v65i0.20894).

Author's response: We acknowledge that our overview of previous studies in this field came across as unfairly brief. The point that we wished to make was that, in a similar vein to this work, Ganesan et al. (2014) treated the solving of hyperparameters as a single-step problem alongside the estimation of emissions in an MCMC framework. The work previous to this, as mentioned by the reviewer, considered the problem as a two-step process, that we believe leads to some difficulties in accurately apportioning uncertainties in a Bayesian framework. We accept that the way this was written made it appear as if only Ganesan et al. (2014) had addressed this problem which was not our intention. We have rewritten this paragraph from page 4 line 31 and extended it to encompass a review of previous work as follows:

"In addition to being dependent on the partitioning of basis functions, Bayesian inversions are also dependent on the form of the PDFs used to describe the prior and likelihood. The terms that describe these PDFs such as the mean, standard deviation and correlation length are commonly referred to as hyperparameters. The dependence of the posterior parameters on these hyperparameters, and a lack of objective determination of their values have been previously identified as a limitation of Bayesian inverse methods (e.g. Rayner et al.,1999). There have since been a number of studies that have proposed methods for determining hyperparameter values using the data (e.g. Michalak et al., 2005, Berchet et al., 2013, Wu et al., 2013). In general, these methods rely on Gaussian assumptions and are performed in a two-step process whereby the hyperparameters are first optimised, and then parameter inference is performed based on these optimal values. Winiarak et al. (2012) also extended this to a semi-Gaussian prior PDF, such that the source term was constrained to be positive. However, as noted by Berchet et al. (2015), one issue is that the uncertainty in the specification of the hyperparameters in step one is not included in the second step. Ganesan et al. (2014) presented an alternative method, where the hyperparameters and parameters were estimated simultaneously using an MCMC algorithm. This framework explored the "uncertainties in the uncertainty", resulting in a more complete characterization of the uncertainty in the posterior parameters. The framework also has the advantage

that the data is used only once, thus remaining strictly Bayesian, and PDFs are able to take forms other than Gaussian. In the transdimensional case, the posterior distribution of the number of unknowns can be heavily dependent on the prescribed uncertainties (Bodin et al. 2012). As such, it is important to incorporate data driven hyperparameters into the transdimensional inversion, if the derived number of unknowns is to be truly dependent on the data."

5. P. 5, l. 25: "its".

Author's response: We thank the reviewer for pointing out this spelling mistake.

6. P. 6, l. 7: "unintelligent" sounds harsh and a softer word would be more appropriate

Author's response: We have replaced "unintelligent" with . . . "unrefined".

7. P. 7, l. 7: the authors should also refer to earlier studies.

Author's response: On reflection this sentence seems redundant, and we have significantly edited this passage in response to the comment below and those of reviewer 2 (see below).

8. P. 7, l. 10: the validity of this assumption should be discussed. At first glance, it looks poor. For instance a large dependency in the case of natural $CO_2$ fluxes over land was shown by Chevallier et al. (2012, doi:10.1029/2010GB003974, their Fig. 5). More generally for instance, it is very likely that hyperparameters are not the same at city-scale and at country-scale.

Author's response: There was an omission in the original manuscript that neglected to mention that this assumption only applies to the dimension of the hyperparameters describing the data, $\theta y$. The hyperparameters describing the prior parameters PDF, $\theta x$, are also dependent on the dimension of the basis functions (i.e. each basis function is described by its own hyperparameters). However, the hyperparameters describing the model-measurement covariance structure are still independent of the number of basis functions, since the prior and data must maintain independence to be strictly

Bayesian. We have rewritten Eq. (8) to account for the fact that there are two distinct sets of hyperparameters, those describing the prior emissions error structure and those describing the model-measurement error structure. We have edited the text in this passage from page 8, line 8, so that it now reads:

"In addition to m and k we also wish to solve for the set of hyperparameters that describe the prior parameters PDF, $\theta$x, and the likelihood PDF, $\theta$y. The dimension of the latter can be assumed independent of k since it is a property of the data. However, we prescribe the dimension of the emissions hyperparameters $\theta$x, to be dependent on k, alongside the parameters. The full form of the transdimensional, hierarchical Bayesian equation then becomes:"

9. P. 9, l. 14: how can the prior location and emissions variables be independent of each other?

Author's response: The prior probability of a Voronoi nucleus occupying any particular position on the grid is dependent only on the total number of Voronoi nuclei, since each nucleus cannot occupy a grid cell that is already occupied. A priori, we assume that there is an equal probability of choosing each grid cell as a nucleus location, and hence this is independent of the emissions. While the magnitude of emissions within each Voronoi cell will be dependent on its location, the prior scaling of this magnitude is independent of the value within it, and thus the location. A priori the scaling of the prior is the same throughout the spatial domain and so the location and emission variables are independent. We note that this condition is only met a priori, and thus significant correlations might be expected on the introduction of the data, and therefore in the posterior distribution. We believe that a reordering of statements may help explain this independence, and have changed the text accordingly. We first stipulate that a uniform distribution is assumed for the location of each nucleus, and that we are solving for a scaling of the underlying emissions distribution. In response to this comment, and a similar one from reviewer 2 we have included the following on page 10, line 21:

"If the emissions value is taken to be some scaling of a prior distribution of emissions then the a priori scaling of the prior emissions field should be one everywhere, and hence this is not dependent on location. In this work we assume a uniform distribution for the location of the Voronoi nuclei, meaning that the prior distribution is independent of the emissions. Given this independence of the variables, the term p(m|k) can be decomposed into two terms expressed as:"

10. P. 13, l. 7 and l. 9: why is there a notion of convergence (l. 9; like if we were looking for just the most-likely state) while the algorithm explores the space of the posterior pdf (l. 7 and 24)?

Author's response: The notion of convergence refers to the convergence in our exploration of the posterior PDF, rather than convergence to a point. The chain starts from one distribution (the prior) and on the introduction of the data moves to another distribution (the posterior). While the individual iterations of the chain will continue to explore the parameter space, the posterior distribution itself should be stationary in order for convergence to be said to have occurred. In light of this comment we have attempted to be clearer about our definition of convergence in the text. Page 16, Line 12 now reads:

"The chain must be run for a sufficient number of iterations in order for convergence of the posterior distribution to occur. The convergence refers to the stability of the distribution across the sampled iterations of the Markov chain."

11. P. 13, l. 19: what does "typically" mean here?

Author's response: Typically here is possibly a misnomer since it will be somewhat dependent on the acceptance rate of the dimension changing proposals in particular. Previous examples of transdimensional inversions in the geosciences (e.g. Bodin and Sambridge 2009, Ray and Key, 2012) have reported running for around one million iterations, although the acceptance rates were fairly low in these studies. Our acceptance rates for the dimension changes are around 30%, although, as mentioned in the

manuscript, this may be due to limited constraint over areas of low emissions such as the sea. The key is that it is important to run the chain for a sufficient number of iterations such that it returns a meaningful stationary estimate of the posterior distribution. Our own tests have shown that $O(10^5)$ iterations are required in order to achieve a robust estimate of the posterior distribution in the particular problems we have attempted. We have altered page 16, line 23 to be more explicit about our meaning:

"In order to achieve a stationary posterior distribution for the parameters, the number of iterations for which the chain. . ."

12. P. 13, l. 27: why should the solution of the problem (independent of the resolution method) be smooth? In other words, is it an advantage or an inconvenient to generate a smooth solution?

Author's response: The word "smooth" was used to mean that the mean of the posterior distribution can provide a spatial distribution that is at a higher resolution than the coarser basis function partitioning at each individual iteration. In this work, since we limited the shape of the Voronoi cells to follow the underlying NAME output grid, the smooth solution is at the resolution of this grid, and therefore still discretized. For inference on national scale fluxes a smooth solution such as this may not be necessary. However, we believe that for the regional or spatial attribution of emissions then a smooth solution is an advantage, since the derived spatial patterns are not dependent on a single partitioning of the basis functions. In light of this comment we have attempted to be clearer about what naturally smoothed means on page 17, line 1:

"a naturally smoothed solution, (i.e. at the resolution of the underlying finite grid) without the need to specify. . ."

13. P. 15, l. 10: is "twice as small" significant here?

Author's response: We do not think that the approximate factor of two is significant per se. It is simply the fact that the RMSE is smaller that is significant. We have edited this

sentence to reflect this point: "The RMSE value of 1.0 ppb was smaller (approximately a half) than that of the subjectively determined grid, for this particular pseudo-data example."

14. P. 17, l. 8: how stable is the estimate with respect to the number of iterations?

Author's response: The UK total is stable with respect to the number of iterations after the 100,000 iteration burn-in period, showing how this distribution has converged. Given the thinning of the chain, for each 100,000 iterations 1000 samples are stored. 500,000 iterations were chosen to allow for the 5 different proposal types at each iteration and after thinning of the chain, the posterior distribution is then estimated by 5000 samples. The first 1250 return a mean UK estimate of 2.27 (2.04-2.48) Tg/yr, the second quarter 2.25 (2.07-2.47) Tg/yr, the third 2.30 (2.04-2.57) Tg/yr and the final 1250 iterations have a mean of 2.27 (2.02-2.53) Tg/yr. This shows how relatively stationary the distribution is with respect to the number of iterations. If this were not the case then either a longer burn-in period may be required, or a change to the proposal jump sizes to allow for more efficient exploration of the chain. In light of this comment we have added the following line to page 21, line 21:

"The UK and Ireland estimates were found to be stable with respect to the number of iterations from which the posterior distribution was sampled. This shows that the burn-in period was sufficient for convergence of these national scale emission totals to occur."

15. P. 18: the first paragraph on the page reminds of the discussion by Berchet et al. (2013, doi:10.5194/acp-13-7115-2013, their sections 3.1, 3.2 and 3.3) on the same topic. This may be acknowledged.

Author's response: In response to a point from reviewer 2 we have added the following section, which also makes reference to Berchet et al. (2013) on page 22, line 9:

"No significant difference was found between the uncertainties derived for times when

local influence was high and those when it was not. By contrast, Berchet et al. (2013) reported CH4 observation uncertainties that were on average 23-31% smaller during the day than at night for a number of sites across Europe using three different hyper-parameter optimization schemes. There are known errors in boundary layer modelling that are likely to be greater at night, although these may be more systematic than random. A better understanding of modelling uncertainties, and how they can be accounted for in the hierarchical framework would be necessary to include this potential bias."

16. P. 18, l. 11: "To avoid this, the. . ."

Author's response: We thank the reviewer for pointing this out and have changed it accordingly in the text.

17. P. 19, l. 6: the sentence is too trivial to be the last one.

Author's response: We have removed this sentence and replaced it with the following: "The framework provides an alternative approach to using a single partitioning of basis functions when performing dimension reduction."

[Figure]

**Supplement:**

[revised manuscript text omitted]

---

## Author Comment (AC2) · 20 Jul 2016

We thank the anonymous reviewer for their comments on the manuscript. We have replied to each of the specific comments in turn below. Page and line numbers given in the author response refer to the marked up version of the manuscript, provided as a supplement to this comment.

Main comments

1. From time to time, a crucial lack of details.

Author's response: We hope we have addressed this general comment in our response to the specific points outlined below.

[Figure]

2. The bibliographical account on hyperparameter estimation is very misleading. As opposed to what is implied in the manuscript, this subject has been addressed in tracer inversion and greenhouse gas inversion for more than 10 years now. I have some knowledge on all the techniques that are addressed by the authors, and I can tell that this poor bibliographical account clashes with the rather good account on the other mathematical aspects.

Author's response: We acknowledge that our discussion of hyperparameter estimation was unduly brief, although we believe that part of the issue may have been in the way it was written. It was not our intention to suggest that the subject was only recently addressed, merely that it was only recently addressed in an MCMC framework as a one-step estimation process, as opposed to the two-step methods advocated elsewhere. We have altered various points of the manuscript where this oversight occurred and address them specifically in response to the points below.

3. Several passages of the manuscript are odd and difficult to understand. This tells me that the manuscript has not been polished enough yet, although it is already quite enjoyable.

Author's response: We hope that the changes made in response to the comments below, and to Reviewer 1's comments have helped to provide a more polished manuscript.

Minor points or comments related to the main points

1. Page 1, line 9, "it allows the uncertainty in our choice of aggregation to be carried through to the solution" is too vague for the abstract. Please clarify or postpone this statement.

Author's response: We have edited this sentence to say: "Therefore, the uncertainty that surrounds the choice of aggregation is accounted for in the posterior parameter distribution."

2. Page 1, Eq.(1): It is common practice to insert equations within the flow of the

article and use punctuation marks at the end of equations (or not if embedded within a sentence).

Author's response: We thank the reviewer for pointing this out. We have attempted to make reference to each equation such that each one's appearance is within the flow of the article. We have further included punctuation marks at the end of the equations.

3. Page 2, line 16: "...or the solution being overly influenced by an incorrect prior, giving the so-called smoothing error": This statement is partially misleading to me. The fact that there are more degrees of freedom is not an issue per se. This has been shown in Bocquet et al. (2011): in theory the more resolved the grid, the better the inversion. It is in addition the fact that the prior could be incorrect that may lead to the smoothing error. If the incorrectness in the prior is low, then such a balance might be pointless. In that case, dimensional reduction is essentially only important for computation issues (which is critical) as pointed out by Bocquet et al. (2011) and ultimately confirmed in Turner and Jacob (2015).

Author's response: We have attempted to be clearer that if the prior is poorly specified then the smoothing error may be important. In addition we have made reference to Bocquet (2011), with regard to the advantage (and disadvantage) of a more highly resolved grid. Page 2, line 22 now reads: "If the prior is poorly specified, this could lead to the solution being overly influenced by the incorrect prior, leading to a so-called smoothing error (e.g. von Clarmann, 2014). If the error in the prior is low, then, while a greater number or degrees of freedom would improve the ability to fit the data, the computational expense of such a calculation may be critical (Bocquet et al. 2011)."

4. Page 2, line 28: "Various studies..." Obviously there has been only a few studies so far. Please mitigate your statement.

Author's reponse: We have edited the beginning of this sentence to say: "There are only a few studies that..."

[Figure]

5. Pages 2, line 33-35: "Although a parameter dimension was successfully identified which minimised the total error, ultimately the choice of model to use was as much influenced by computational efficiency, as it was by this combination of aggregation and smoothing error": Yes, just as predicted in Bocquet et al. (2011). This could be mentioned.

Author's response: We have included the following line in addition to the above sentence on Page 3, line 12: "This follows the work of Bocquet et al. (2011) who showed that the highest resolution grid should have the smallest total error, and thus computational efficiency is the main driver behind dimension reduction."

6. Page 3, line 2-3: "Therefore, the uncertainties in step one do not necessarily propagate through to step two." All of the objective criteria in Bocquet et al. (2011) depend on the observation network. One of the criterion in Bocquet et al. (2011) actually depend on the data itself (section 4.1.3 and illustrated on Fig. 5 of the same paper).

Author's response: We do not deny that the criteria in Bocquet et al. (2011) are dependent on the observation network. The point that we wished to make is that although the criteria are objective, ultimately there is one optimal grid that is used. This choice is dependent on the transport model, prior and covariance structures and so uncertainties in these should propagate through to the choice of basis functions. In using one optimal grid, the uncertainty that surrounds this choice of grid is not then carried through to inference on the parameters of interest.

With regard to the observation-dependent criterion, Bocquet et al. (2011) admit that an "inversion crime" is committed, in that the same data used to construct the optimal grid is then used to perform Bayesian inference on the set of parameters under investigation. The approach of our work overcomes this issue, and that of incorporating the uncertainty surrounding the choice of grid, by simultaneously using the data to perform inference on the basis function dimensionality as well as the parameters of interest.

7. Page 4, lines 20-30: This paragraph gives a wrong and totally biased picture of

the literature on the hyperparameter estimation as used in tracer/greenhouse gas inversions, not to mention geophysical data assimilation and in particular meteorology. There are dozens of papers on the subject before the contributions of Ganesan et al. Only focusing on tracer inversions, one of the very first use for the inversion of the Chernobyl source term is in Davoine and Bocquet (2007) which has been extended to non-Gaussian inversion problems in the Fukushima case (Winiarek et al., 2012). But there really are dozens of papers on the subject. Two reviews on the matter are Michalak et al. (2005) and Wu et al. (2013). It would be fair to mention those papers before mentioning Ganesan et al.

Author's response: We apologise if the discussion of hyperparameter estimation came across as unduly brief and focused on one work. Of course there have been many previous studies on hyperparameter estimation, and it was not our intention to suggest otherwise. We believe that the phrasing of the first sentence in this paragraph may have given the impression that the work of Ganesan et al. (2014) was the only study to look at this. What was meant was that the novelty of Ganesan et al. (2014) was in the hierarchical framework, where the hyperparameters were solved simultaneously with the fluxes. The reversible jump algorithm used in this work is a more general version of the MCMC framework used in Ganesan et al. (2014). As such, the transdimensional framework is a natural extension of this previous hierarchical framework; hence the focus on this related approach. However, we acknowledge that a more complete picture of other approaches to this problem may be required. The issue of defining the hyperparameters has been identified in numerous studies, and addressed in various others as mentioned by the reviewer. We have rewritten this paragraph of the text to reflect this fact. Page 4, line 31 now reads:

"In addition to being dependent on the partitioning of basis functions, Bayesian inversions are also dependent on the form of the PDFs used to describe the prior and likelihood. The terms that describe these PDFs such as the mean, standard deviation and correlation length are commonly referred to as hyperparameters. The dependence

of the posterior parameters on these hyperparameters, and a lack of objective deter-
mination of their values have been previously identified as a limitation of Bayesian
inverse methods (e.g. Rayner 1999). There have since been a number of studies that
have proposed methods for determining hyperparameter values using the data (e.g.
Michalak 2005, Berchet 2013, Wu 2013). In general these methods rely on Gaussian
assumptions and are performed in a two-step process whereby the hyperparameters
are first optimised, and then parameter inference is performed based on these optimal
values. Winiarak et al. (2012) also extended this to a semi-Gaussian prior PDF, such
that the source term was constrained to be positive. However, as noted by Berchet et
al. (2015), one issue is that the uncertainty in the specification of the hyperparame-
ters in step one is not propagated through to the second step. Ganesan et al. (2014)
presented an alternative method, where the hyperparameters and parameters were
estimated simultaneously using an MCMC algorithm. This framework explored the
"uncertainties in the uncertainty", resulting in a more complete characterization of the
uncertainty in the posterior parameters. The framework also has the advantage that
the data is used only once, thus remaining strictly Bayesian, and PDFs are able to take
forms other than Gaussian. In the transdimensional case, the posterior distribution of
the number of unknowns can be heavily dependent on the prescribed uncertainties
(Bodin et al. 2012). As such, it is important to incorporate data driven hyperparam-
eters into the transdimensional inversion, if the derived number of unknowns is to be
truly dependent on the data."

8. Page 5, whole section 2.1: this subsection 2.1 is totally off in the flow of the paper,
especially starting from line 10. I do not understand why the past tense is used. I do not
understand which numerical experiment you are referring to? This must be re-written
or postponed in the manuscript.

Author's response: We thank the reviewer for pointing out that this section appears
out of sync with the flow of the paper. We had felt it important to introduce the model
early on in the paper so that when referring to the sensitivity matrix, H, it would be

clearer what this represented. However, given that the method is general, and the use of NAME specific, we agree that this section would be better suited at a later stage of the manuscript. This section now appears on page 17 after the discussion of the more general reversible-jump method. We have further edited parts of this section to clarify the use of NAME for this application. In addition, we have added a definition of the linear forward model into the introduction, so that the relationship between observations, model and emissions is made clear from the start. Page 2, line 1 is given by:

"The relationship between the observations and parameters can be determined by a CTM. For flux estimation problems this forward model is usually given by the linear relationship: $y = Hx + \varepsilon$ (2) Where H is a matrix of sensitivities of the observed mole fractions to a change in emissions from a finite grid, calculated by the CTM, and $\varepsilon$ represents random representation errors of the observations."

9. Page 5, section 2.1: The use of a Lagrangian model such as NAME adds further interesting issues that were discussed in Koohkan et al. (2012). There is an additional uncertainty due the number of particles, especially when just a few of them fall into grid-cells. This issue could conflict with the transdimensional approach.

Author's response: We acknowledge that there will be some uncertainty regarding the number of particles required in a Lagrangian simulation as discussed by Koohkan et al. (2012). However, it is not immediately clear how this would conflict with the transdimensional approach, any more than it would conflict with a fixed grid approach. The output of the NAME model is dominated by the mean plume of the back-trajectory which may be well represented using a release rate of around 1000 particles per hour. Manning et al. (2011) reported that release rates greater than 11,000 particles per hour in NAME had a negligible impact on noise within the inversion domain. In this work, particles were released at a rate of 20,000 per hour, so under-sampling should not be a significant issue, particularly in the areas of interest, which are close to the measurement sites. The 30 day back-trajectory of each 2 hour release period is thus estimated by 40,000 particles. As a result of this comment we have added the particle

release rate to this section.

10. Page 5, line 25: "Furthermore, each grid cell within each aggregated region has an enforced correlation to it's neighbours": Why?

Author's response: Grid cells of the same aggregated region must take the same value by the very definition of the aggregation, hence the enforced uncertainty correlation. This sentence omitted the word "uncertainty" which we have now included: "Each grid cell within each aggregated region has an enforced uncertainty correlation to its neighbours..."

11. Page 5, line 25: "it's" $\longrightarrow$ "its". Author's response: Corrected.

12. Page 6, lines 19-24: "In this work, we approximate the form of the Voronoi cells, by restricting them to those points on the underlying finite grid which are closest to their respective nuclei. As such, the region edges are not exactly equidistant between nuclei, but this approach was taken since the exact form of the Voronoi cells is unimportant, and each underlying grid cell belongs to only one nuclei, making computation very simple." This passage is very unclear to me. Please give more details.

Author's response: The true Voronoi tessellation would result in all points within a Voronoi cell being closer to that cell's nucleus than any other. This could result in the native resolution grid cells from the CTM being split between regions. This would require some additional calculation of the proportion of each grid cell within each region. For simplicity, and since it does not seem appropriate to attempt to resolve emissions at a resolution greater than that of the CTM output, we therefore restrict the borders of each Voronoi cell to follow the regular underlying grid. We have rewritten this passage on Page 7, line 16 to say:

"For simplicity, in this work we define the form of the Voronoi cells such that they must be composed of whole grid cells of an underlying finite grid. As such the whole of each native resolution grid cell can belong to only one Voronoi region, and grid cells are

not split between two or more regions. This approach limits the maximum resolvable resolution to be the same as the underlying grid (defined by the CTM output)."

13. Page 6, line 7: "as shown by Ganesan et al. (2014)." You are pushing the envelope too far here. This was well known and shown a long time before Ganesan et al. (2014). Please remove this statement which is biased.

Author's response: We have removed this sentence as on reflection it is somewhat redundant. We have further rewritten this paragraph on in response to the comment below. Page 8, line 8 now reads:

"In addition to m and k we also wish to solve for the set of hyperparameters that describe the prior parameters PDF, $\theta$x, and the likelihood PDF, $\theta$y. The dimension of the latter can be assumed independent of k since it is a property of the data. However we prescribe the dimension of the emissions hyperparameters, $\theta$x, to be dependent on k, alongside the parameters. The full form of the transdimensional, hierarchical Bayesian equation then becomes:"

14. Page 6, line 10: "Where the assumption has been made that $\theta$ is independent of k": This is a question that has puzzled me for a long time. It may very well be that this independence is plain wrong and that it has a strong impact on the resulting inversions. At the very least, you should discuss that assumption.

Author's response: There was a crucial omission in the original manuscript that should help clarify this assumption. The hyperparameters vector, $\theta$, can be split into two components: $\theta$x describing the prior emissions PDF, and $\theta$y which describes the model-measurement covariance structure. $\theta$y should be independent of the basis function dimension since this describes uncertainties in the data. However, there was a mistake in that we failed to mention that $\theta$x are also dependent on the dimension of the basis functions, so that each basis function has its own unique hyperparameters.

We have rewritten Eq. (8) to account for the fact that there are two distinct sets of

hyperparameters, those describing the prior emissions error structure and those describing the model-measurement error structure. Equations (18) and (19) have also been rewritten to account for the fact these describe a multivariate PDF.

15. Page 6, lines 10-20: "Whereas the hierarchical framework alone can be solved through conventional Markov Chain - Monte Carlo (MCMC) methods (Ganesan et al., 2014), since the dimension of m is variable in the transdimensional case, it must be solved by a different, though strongly related, approach." The sentence is unclear. Please rephrase.

Author's response: We have changed this to say on page 8, line 16: "Ganesan et al (2014) used the Metropolis-Hastings algorithm to simultaneously solve for $\theta$ and x in the hierarchical framework. In the transdimensional inversion we also wish to vary the dimension of m, which can be achieved using the more general rj-MCMC technique described by Green (1995), as set out in the following section."

16. Page 8, line 15, Eq.(11): n $-\to$ n.

Author's response: Corrected.

17. Page 9, line 2: hyperparameter $-\to$ hyperparameter.

Author's response: Corrected.

18. Page 9, line 3: "The other three proposals" $-\to$ "The other three proposal ratios".

Author's response: We are not convinced that this change is appropriate since it is the type of proposal (birth/death/move) that is referred to, rather than the ratio of the new state to the current state. However, to make this clearer in the text we have changed this sentence to: "The other three proposal types involve . . ."

19. Page 9, lines 4-5: "In effect, this means a change in the sensitivity matrix, H, that maps the relationship between emissions and observations.": Please be more much precise as to what it implies for H.

Author's response: We have changed page 10, line 12 to: "This means that the sensitivity matrix, H, which maps the relationship between emissions and observations, must be recalculated for the new set of aggregated regions."

20. Page 9, line 9, "there are an unknown number of unknowns" −→ "there is an unknown number of unknowns"?

Author's response: Corrected.

21. Page 9, line 10, "be decomposed to two separate terms" −→ "be decomposed into two separate terms".

Author's response: Corrected.

22. Page 9, lines 10-11: "since the prior location and emissions variables are independent of each other" is not obvious to me. Can you please elaborate?

Author's response: The prior probability of a Voronoi nucleus occupying any particular position on the grid is dependent only on the total number of Voronoi nuclei. This is because each nucleus cannot occupy a grid cell that is already occupied and a priori, we assume that there is an equal probability of choosing each grid cell. While the distribution and magnitude of emissions within each Voronoi cell will be dependent on its location, the prior scaling of this magnitude is independent of the value within it, and thus the location. A priori the scaling of the prior is the same throughout the spatial domain and so the location and emission variables are independent. We note that this condition is only met a priori, and thus significant correlations might be expected on the introduction of the data, and in the posterior distribution. In light of this comment and those from reviewer 1 we have included the following on page 10 line 21:

"If the emissions value is taken to be some scaling of the a priori distribution of emissions, then the a priori scaling of the prior emissions field should be one everywhere, and hence this is not dependent on location. In this work we assume a uniform distribution for the location of the Voronoi nuclei, meaning that the prior distribution is inde-

pendent of the emissions. Given this independence of the variables, the term p(m|k) can be decomposed into two terms expressed as:"

23. Page 9, lines 19: "so they may be located anywhere" $-\rightarrow$ "so they may be located a priori anywhere".

Author's response: Corrected.

24. Page 9, line 20: "can be located on the finite underlying grid": that is an imprecise statement. Could you please be more specific?

Author's response: Page 11, line 6 now reads: "If we assume that the Voronoi nuclei can only be located at the centre points of each grid cell on a finite underlying grid with K grid cells, and that no two nuclei can occupy the same grid cell..."

25. Page 9, Eq.(16): Even using an – as much as possible – uninformative prior may slightly influence the number of nuclei. Can you elaborate on that?

Author's response: This is true, in so much as the bounds of the uniform distribution may be restrictive if set too narrow, or if the constraint from the data is weak. As a result of this comment we have added the following page 11, line 13:

"Whilst the uniform prior is relatively uninformative, the choice of maximum and minimum bounds may still influence the number of nuclei if the constraint from the data is weak or if the bounds are too narrow."

26. Page 10, Eq.(18): Shouldn't x be bold?

Author's response: Yes, we thank the reviewer for pointing this out.

27. Page 11, line 17: "and the other a correlation length between measurements, $\tau$ .": That is why I have reservation on the fact that all of the hyperparameters are independent from k.

Author's response: It is only the a priori values of the hyperparameters that are considered independent of k. On taking into account the effect of the data, correlations between these values will develop. We also note that the initial phrase should have said the dimension of $\theta y$ is independent of k, which will continue to hold in the posterior since this dimension does not change in the inversion.

28. Page 11, line 18: "it's" $-\rightarrow$ "its".

Author's response: Corrected.

29. Page 11, line 18-21: Please elaborate. The statements are too concise. Author's response: We have expanded on these statements by including a description of the covariance structure and how this leads to a simplified calculation of the inverse and determinant on page 13, line 20 – page 14, line 13. Please refer to the marked up version of the manuscript provided as a supplement to this comment to see the changes and equations now included in this section.

30. Page 12, line 19: "it's" $-\rightarrow$ "its".

Author's response: Corrected.

31. Page 12, line 24-27: What is |J| here?

Author's response: The Jacobian that accounts for the scale changes is 1, as mentioned on Page 9, lines 14-19.

32. Page 12, line 25: "In practice, this means that one does not have to define the nuclei locations as being restricted to the locations of the underlying grid, and they can in fact take any position within the inversion domain." Okay, but what did you do in this study?

Author's response: In this work we still defined the nuclei as restricted to the centre points of the grid of the NAME output. This was done since it would be less computationally efficient to interpolate sensitivity values at resolutions finer than the output resolution of NAME. Increasing the resolution beyond that of the meteorological fields

used to generate the output may also give rise to further modelling errors. In light of this comment we have added the following sentence to the page 15, line 25:

"However, since it makes little sense to solve at a resolution finer than the native resolution of the sensitivity maps generated by the CTM, in this work we continued to restrict the nuclei locations to the centre points of the underlying grid."

33. Page 13, line 9: "in order for convergence to occur" $-\rightarrow$ "in order for the convergence to occur".

Author's response: In response to reviewer 1's comments we have altered this sentence slightly, and thus we do not feel the inclusion of "the" is necessary.

34. Page 14, line 1: Can you describe the temporal dimension of the emissions. For instance, are they modulated in time?

Author's response: The EDGAR inventory provides an annual estimate of anthropogenic emissions. This means that for the inversion timescales considered here (2 months for the pseudo example and 1 month for the real data) the a priori emissions are constant in time. In light of this comment we have included the following line on page 17, line 26:

"This time-independent field was regridded from the native resolution of 0.1x0.1 to the coarser NAME output resolution of 0.234x0.352."

35. Page 14, line 10: How many observations do you use? How long is the time frame?

Author's response: We have edited this line to include these details on the number of observations and the time frame of the pseudo-data inversion. Page 18, line 4 now includes: ". . .using 6-hourly averaged NAME sensitivities from a two-month period May-June 2014. This gave a total of 942 pseudo observations from the four sites, the locations of which. . ."

36. Page 14, line 16: Which first guess (mean prior) did you choose? In general the

reader is missing quite a few details to fully understand the experiment. Please give more information.

Author's response: We acknowledge that this section was in general a little short on key details. We have included further information on the prior chosen, and along with the additional information of the previous two points, hope that this will help the reader to more fully understand the experiment. Page 18, line 12 now reads:

"The initial a priori scaling was 1 throughout the domain, compared to the true chequer-board pattern which had values of 1.5 and 0.5 in the regions of high and low scaling respectively."

37. Page 16, line 30-32: What would happen without this filter?

Author's response: In this example the difference is minimal. For example, at Mace Head the mean derived uncertainties for times of low local influence was 15.8 ppb, while for times of high local influence it was 18.0 ppb. In an inversion performed without the filter the derived UK posterior emissions distribution was almost exactly the same: 2.29 (2.04-2.52) Tg yr-1, as opposed to 2.28 (2.05-2.52) Tg yr-1 when the filter was included. Similarly, we found little difference in the posterior uncertainties when separate values were resolved for day (12:00-16:00) and night (16:00-12:00). We postulate that this is due to the treatment of the uncertainties as random rather than systematic. It is probable that model errors associated with sub-grid scale processes or boundary layer parametizations are systematic, and cannot be well accounted for by the treatment of uncertainties as random in this particular inversion setup. In light of this comment we have included the following on page 22, line 9-14, when discussing the posterior model-measurement uncertainties:

"No significant difference was found between the uncertainties derived for times when local influence was high and those when it was not. By contrast, Berchet et al. (2013) reported CH4 observation uncertainties that were on average 23-31% smaller during the day than at night for a number of sites across Europe using three different hyperparameter optimization schemes. Errors in boundary layer modelling are likely to be greater at night, although these may be more systematic than random. A better understanding of modelling uncertainties, and how they can be accounted for in the hierarchical framework would be necessary to include this potential bias." There was an error in the original manuscript in that the threshold should have said 30% not 40%. This has been corrected.

38. Page 17, line 1-4: Please provide a figure.

Author's response: We have included a figure of the six fixed regions and the sub-domain in which the regions were variable in the inversion, and make reference to this figure in this paragraph (Figure 6).

supplement.pdf

—————————————————————

[Figure]

**Supplement:**

[revised manuscript text omitted]